# *Streptomyces*-Derived Bioactive Pigments: Ecofriendly Source of Bioactive Compounds

Aixa A. Sarmiento-Tovar [1,2], Laura Silva [1,2], Jeysson Sánchez-Suárez [2] and Luis Diaz [2,3,*]

1    Master Program in Design and Process Management, School of Engineering, Universidad de La Sabana, Chía 140013, Colombia
2    Bioprospecting Research Group, School of Engineering, Universidad de La Sabana, Chía 140013, Colombia
3    Doctoral Program of Biosciences, School of Engineering, Universidad de La Sabana, Chía 140013, Colombia
*    Correspondence: luis.diaz1@unisabana.edu.co

**Abstract:** Pigments have been used since historical times and are currently used in food, cosmetic, pharmaceutical, and other industries. One of the main sources of natural pigments are plants and insects; however, microorganisms are of great interest due to their bioactivities and advantages in their production. Actinobacteria, especially the genus *Streptomyces*, are biotechnologically valuable, producing specialized metabolites with a broad spectrum of bioactivities, such as antioxidant, anti-cancer, antibiofilm, antifouling, and antibiotic activities, as well as pigments, among others. In this review, we identify, summarize, and evaluate the evidence regarding the potential of *Streptomyces* strains to be biological sources of bioactive pigments. To conclude, future research will include purifying pigmented extracts that have already been reported, studying the purified compounds in a specific application, isolating new microorganisms from new isolation sources, improving the production of pigments already identified, modifying culture media or using new technologies, and developing new extraction techniques and a wide range of solvents that are ecofriendly and efficient.

**Keywords:** pigment; *Streptomyces*; antioxidant activity; antimicrobial activity; cytotoxic activity; antifouling potential; antibiofilm potential

## 1. Introduction

Pigments are used to manufacture various products because they can enhance the natural color or replace color lost during the manufacturing process, generating greater consumer appeal by adding a novel sensory aspect. Since the introduction of synthetic dyes by Perkin in 1856 [1], their production has increased, and natural colorants from plants and animals have decreased due to synthetic pigments being relatively cheaper [2].

In the 20th century, natural organic pigments were almost entirely displaced by synthetic molecules such as phthalocyanines, ranging from blue to green, and quinacridones, ranging from orange to violet [3]. Advances in organic chemistry allowed these compounds' mass production to replace the natural ones, which are often more complex to acquire [4]. Therefore, the use of synthetic organic dyes has been the most cost-effective approach for years [5]. Synthetic dyes are superior to natural pigments in their staining power, ease of application, stability, and cost/effect [6,7].

The production of synthetic pigments in 2019 represents $8 \times 10^5$ tons per year [6,8]; however, some of them have negative effects in the human health and environment [6]. Firstly, they are not biodegradable and are very difficult to remove from effluents or to dispose of due to their stability to oxidation and reduction processes; it is even worse in the case of degradation, their byproducts have been directly or indirectly proven to be health hazards. For example, anionic dye removal is a huge challenge as they produce very bright colors in water and show acidic properties [1,9]. Secondly, the synthetic pigments can affect human health, having negative effects on several vital organs, such as the brain, kidneys, liver, and heart; and systems such as respiratory, immune, or reproductive [6]. Especially,

cationic dyes can cause critical diseases, such as cyanosis, jaundice, quadriplegia, Heinz body formation, and tissue necrosis in humans. Additionally, 130 of 3200 azo dyes in use can form carcinogenic aromatic amines during the degradation process. [1,9,10]. Other secondary effects of synthetic pigments are asthma, allergies, nausea, skin and eye irritation, dermatitis, cancer, hemorrhages, gene mutations, and heart disease [6,10]. Additionally, since 1975 the FDA (Food and Drug Administration) has conducted toxicological studies on synthetic food dyes, finding different irregularities, namely the lack of statistical reliability in the number of animals used and inadequate dosing [11]; even dyes such as red 40 or Allura, which promote tumor formation, have been listed as approved pigments by the FDA [12].

These controversies between synthetic dyes and natural pigments, and the fact that consumers do not accept the first [1], have contributed to the growing interest in recent years in natural dyes, mainly in the food and cosmetic industry [13,14]. In this way, it also contributes to the worldwide trend of replacing synthetic dyes with natural pigments. The pigments extracted from plants or microorganisms imply a certain degree of safety. Due to historical antecedents and consumption patterns, toxicological problems are not as marked as their synthetic counterparts [15]. The above represents that the natural pigment market, only in food industry, is predicted to reach USD 3.5 billion at 12.4 CAGR by 2027 [16].

Natural pigments can be obtained from three main sources: animals, plants, and microorganisms [17]. Although there are many natural pigments, only a few are available in quantities suitable for industrial production [18,19]. Microbial pigments are of great interest due to their stability and culture technology availability [20,21]. The benefits of pigment production from microorganisms include easy and fast growth in economic culture media, independence from climatic conditions, and different colors and shades [22–24]. Thus, microbial pigment production is now one of the promising and emerging fields of research, revealing its potential for various industrial applications [17–19,25–27]. Additionally, some microbial pigments have been reported to possess anticancer activity, contain pro-vitamin A, and have some important properties such as stability to light, heat, and pH [28]. However, from an industrial point of view, developing a high-tech and cost-effective harnessing for the large-scale production of various microbial pigments is necessary [19].

The use of natural pigments in food, textiles cosmetics, and pharmaceuticals has increased in recent decades [1]. Especially, the inclusion of microbial pigments in foods is in response to increased consumer demand for safer and more natural foods [29]. Among these are anthocyanins and betalains, which are used as water-soluble pigments, chlorophylls, and fat-soluble carotenoids [30]. Food-grade fermentative pigments, such as β-carotene and phycocyanin, are currently commercialized [31], and pigments such as indigoids, anthraquinones, and naphthoquinones currently have potential applications in the food industry [32].

Pigmented secondary metabolites include astaxanthin, canthaxanthin, carotenoids, melanins, indigoidine, flavins, and quinones [33], which have demonstrated the efficacy and potential clinical applications in the treatment of various diseases and have certain properties as antibiotic, anticancer, and immunosuppressive compounds [34]. Microbial anthocyanins, for example, are involved in a wide range of biological activities, such as reducing the risk of cancer, reducing inflammatory aggression, and modulating the immune response [35].

In the cosmetic industry, currently, microbial pigments such as prodigiosin and violacein from *S. marcescens* and *C. violaceum* are used commercially for sunscreen application due to its antibacterial and antioxidant capacity, which is similar to ascorbic acid [36]. Moreover, some microorganism, such as *Arthrobacter agilis, Arthrobacter psychrochitiniphilus Zobellia laminarie,* and *Synechocystis pevalekii*, produce UV-protective pigments that can withstand the UV-B and -C radiation, protecting the skin [37–39]. In addition, the most important pigment, melanin, with its important role in protecting the skin, is produced by diverse microorganisms, including *Aspergillus fumigates*, *Vibrio cholerae*, *Cryptococcus*

*neoformans*, *Colletotrichum lagenarium*, *Alteromonas nigrifaciens*, and most of the *Streptomyces* species [23,40].

On the other hand, characterized pigments from *Vibrio* spp (prodigiosin) [41], *Serratia marcescens* [42], and *Janthinobacterium lividum* [43] have been evaluated in the staining of different fibers, including wool, nylon, acrylics, silk, cotton, and polyester microfiber, obtaining good color shades. In addition, due to their antibacterial activity, they are being used in the development of antimicrobial textiles for hospital infections [44].

Among microbial pigments, one of the most interesting genera is *Streptomyces*, due to its great reproductive capacity, and also because one of the most produced pigments in the industry, melanin, can be produced by this bacterium [40,45]. In addition, this type of actinomycetes has a fascinating genetic distribution, which is attractive for replication in the biotechnology industry [46–48]. In addition, *Streptomyces* are well known for their abundant secondary metabolism, which has provided different bioactive compounds, namely antibiotics, anti-inflammatories, antioxidants, and cytotoxins [49–51]. Several of these compounds are colored [52] and, given the bioactivity potential shown for *Streptomyces* strains [23], many of the colored *Streptomyces*-derived compounds could signify an exciting opportunity to find bioactive pigments.

Considering that the need for safe pigments is applicable in different areas, and with the additional beneficial activities and the biotechnological potential of *Streptomyces*, we accomplished a literature review of pigments produced by *Streptomyces*. We identified which ones have some bioactivity such as antimicrobial, antioxidant and cytotoxic activities, which are relevant for determining possible future applications. Afterward, we summarize the conditions to optimize the pigment production of the strains on which this study was performed. This review aimed to identify, summarize, and evaluate the evidence regarding the potential of *Streptomyces* strains as a biological source of bioactive pigments.

## 2. Results

### 2.1. General Findings

The literature search identified 3904 articles, of which 176 were not original and 1253 were duplicates, giving a total of 2475 articles. These articles were screened by reading titles and abstracts, following inclusion or exclusion criteria. From this stage, 112 papers were selected for full-text evaluation. Finally, 53 articles were selected for full-text assessment and were used for data extraction (Figure 1).

Even though the study of *Streptomyces*-derived pigments dates back many years, starting in 1973, most articles (41.7%) were published between 2018 and 2022 (Figure 2a). Thus, it is evident the relevance that the production and evaluation of *Streptomyces*-derived pigments have taken, given the growth in the number of articles published in recent years (Figure 2b).

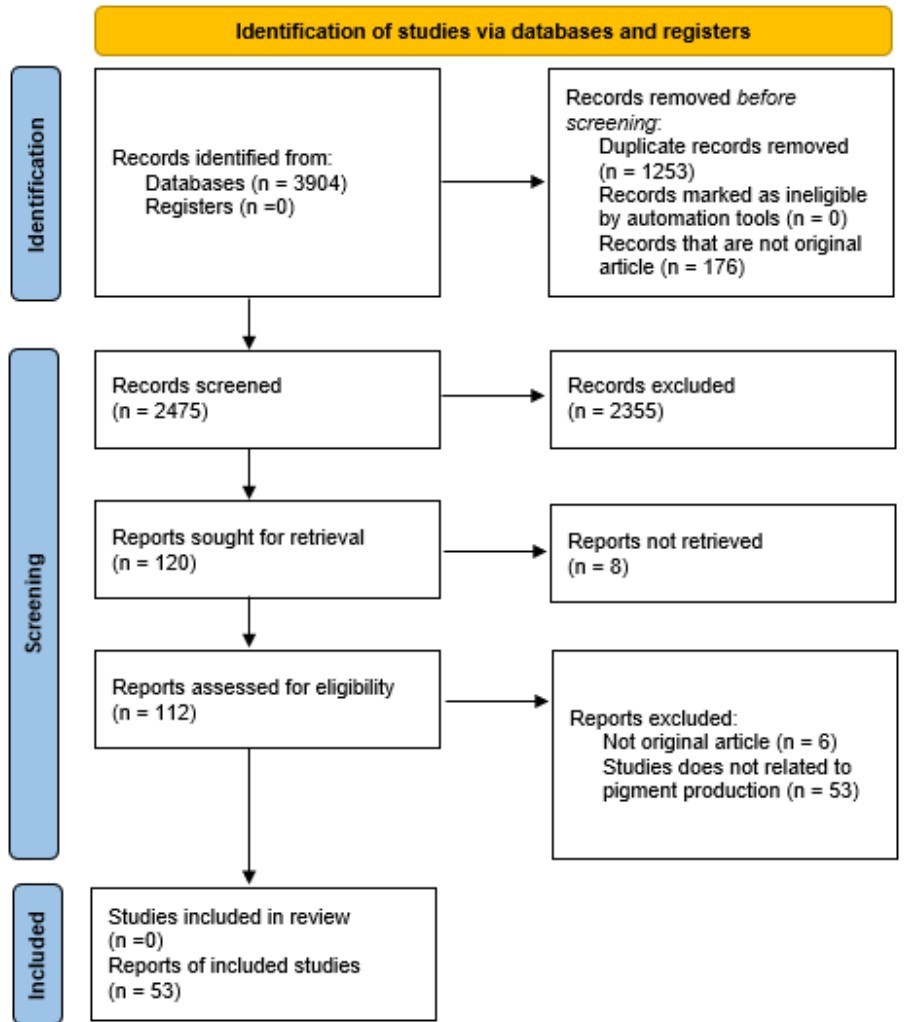

**Figure 1.** PRISMA flow diagram. Flowchart of systematic literature search according to PRISMA guidelines. Modified from: Page, M.J., McKenzie, J.E., Bossuyt, P.M., Boutron, I., Hoffmann, T.C., Mulrow, C.D. et al. The PRISMA 2020 statement: An updated guideline for reporting systematic reviews. (2021) *PLOS Med.* **2021**, *18*, e1003583. doi:10.1371/journal.pmed.1003583 [53] (see Table S1).

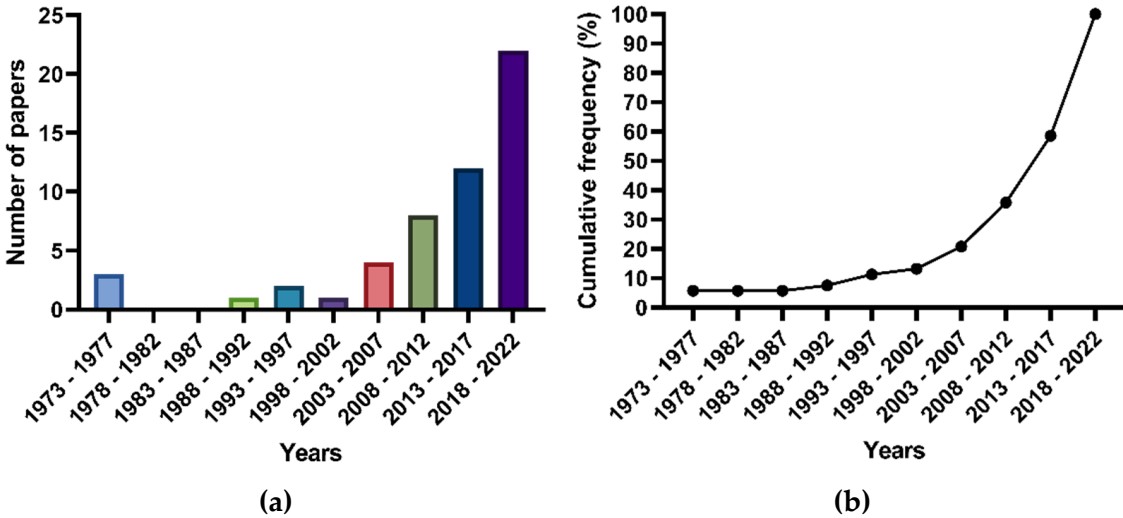

**Figure 2.** General findings around the journals of the selected articles. (**a**) Publication distribution over the time. (**b**) Cumulative frequency (%) distribution.

This increase in the number of items may be due to several reasons: First, consumers demand natural pigments as they are considered safe, nontoxic, noncarcinogenic and biodegradable [1]. Second, the pigment market's exponential growth, which represents USD 36.4 billion in 2021, is projected to expand at a compound annual growth rate (CAGR) of 5.2% from 2022 to 2030 [54]. Third, actinobacteria are among the most profitable and biotechnologically valuable [46,55]. *Streptomyces* especially is responsible for the vast majority of specialized metabolites [46–48,56]. Fourth, the need to investigate unexplored or underexploited habitats as new sources of specialized metabolites [46].

On the other hand, analyzing the map of the countries of the corresponding authors of the articles (Figure 3) showed India and China are the countries with the highest scientific production, while the contribution of articles from Latin America, Europe, and Africa is extremely limited or almost null. In addition, with respect to collection and isolation areas (Figure 4), 20.4% of the articles did not specify the isolation area. India stands out as one of the countries with the highest number of *Streptomyces* collection areas. In this way, Asia is the main continent where the scientific production and collection sites around *Streptomyces*-derived pigments are concentrated.

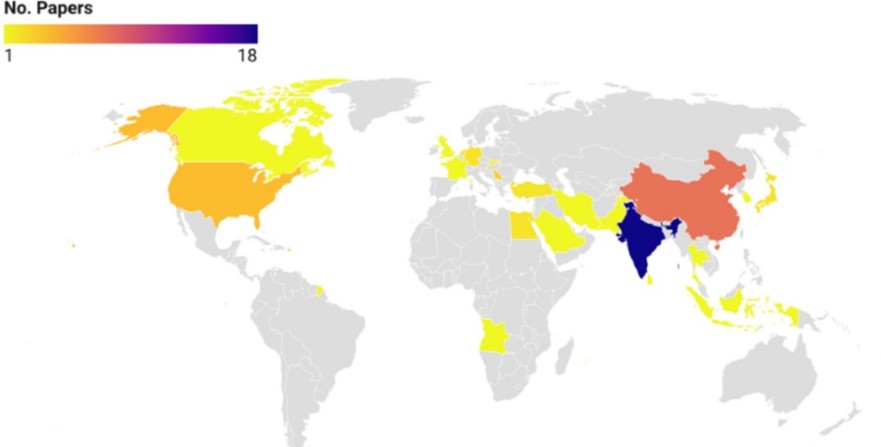

**Figure 3.** World map showing where the articles included in this review were produced, the corresponding author affiliation.

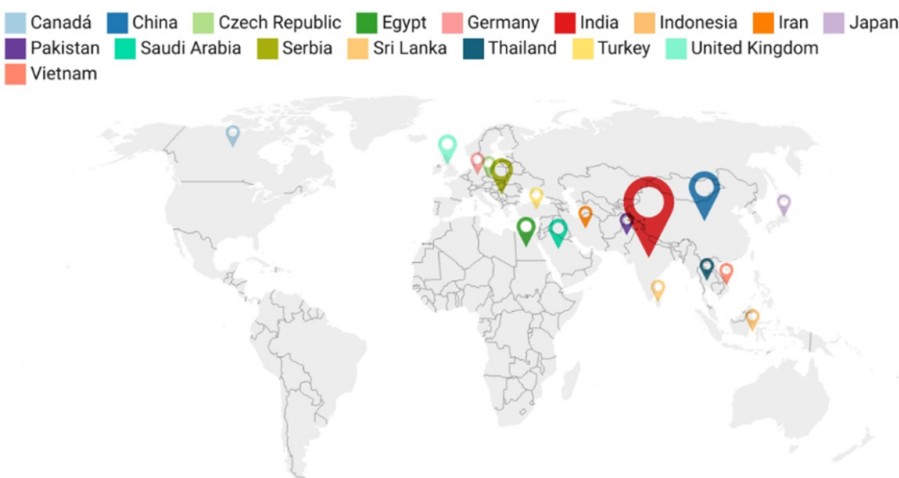

**Figure 4.** World map showing the countries where the *Streptomyces* strains were isolated.

This may be because Asian countries have numerous exotic places to isolate new species of microorganisms [57]; such an example are the tropical forests of Southeast Asia—the reefs of the 'coral triangle' and the river basins are unique on Earth [58]. Specifically, India has the Western Ghats, which are one of the thirty-four biodiversity hotspots

in the world [59], and interesting ecosystems for microbiologists, such as the Vellar Estuary [60], the Gulf of Mannar Biosphere Reserve [61], the Thar Desert [62], and the Sabarimalai forest [63], among others [57].

Another important aspect that was observed in the literature review is the type of substance with which the articles worked. Most of them (50.9%) reported the identification of the pure compound or a partial purification; 43.4% indicated that they evaluated the extract, and a minor quantity worked with fraction and pure cultures (Figure 5a). According to the type of source, it was found that the most used solvent for extraction and purification of the extract was ethyl acetate; other solvents used were methanol, chloroform, and ethanol. As for the source of isolation, 52.8% belonged to soil (Free-living), 7.5% to marine (Free-living) and marine symbionts, and a minor quantity (3.8%) were terrestrial symbionts and freshwater (Free-living). In addition, 24.6% of the articles did not report the source of isolation (Figure 5b).

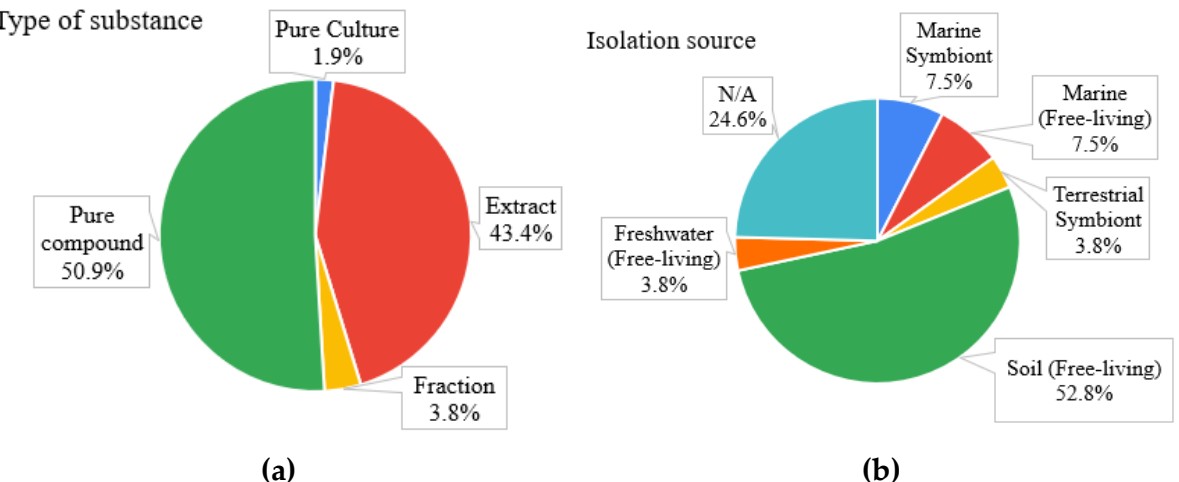

**(a)** **(b)**

**Figure 5.** Characteristics of the information registered in the selected articles. (**a**) Percentage of articles that evaluated either crude extracts (see Table S5), pure compounds (see Table S6), fraction, or pure culture. (**b**) Percentage of articles by the environment where the isolation occurred.

In this way, it is important to keep in mind for future research to purify pigmented extracts that have already been reported, study the purified compounds in a specific application, and isolate new microorganisms from new isolation sources. In addition, it is necessary to study extraction with different solvents and to propose new extraction techniques that are environmentally friendly and have high yields.

Analyzing the number of pigment-producing strains from different isolation sources and the evaluated bioactivities, it was observed that the highest number of reported strains are from the soil (Free-living), with 29 reported strains, and the most evaluated bioactivity was antimicrobial activity. In addition, 10 pigment-producing strains were evaluated for more than 1 bioactivity. On the other hand, the source of isolation that least reported pigment-producing strains is terrestrial symbiont, and 21 of the pigment-producing strains were not evaluated for any bioactivity (see Table 1).

**Table 1.** Number of strains and compounds by bioactivity and type of source.

| Type of Source | Bioactivity | No. Strains | No. Compounds | Ref. |
|---|---|---|---|---|
| Freshwater (Free-living) | Antimicrobial | 6 | 3 | [64,65] |
| Marine (Free-living) | Antimicrobial | 4 | 1 | [60,66] |
| | Antioxidant | 1 | 1 | [67] |
| | Multiple [1] | 1 | N/A [2] | [68] |
| Marine Symbiont | Antimicrobial | 1 | N/A [2] | [69] |
| | Cytotoxic | 1 | N/A [2] | [61] |
| | Multiple [1] | 1 | N/A [2] | [70] |
| | N/A [3] | 1 | N/A [2] | [71] |
| Terrestrial Symbiont | N/A [3] | 1 | 3 | [72] |
| Soil (Free-living) | Antimicrobial | 11 | 3 | [62,63,73–81] |
| | Antioxidant | 1 | 1 | [82] |
| | Cytotoxic | 5 | 2 | [83–87] |
| | Multiple [1] | 8 | 6 | [40,88–94] |
| | N/A [3] | 4 | 1 | [95–97] |
| N/A [1] | Antimicrobial | 4 | 6 | [98–101] |
| | Antioxidant | 1 | 1 | [102] |
| | N/A [3] | 15 | 7 | [103–110] |

[1] More than one of the bioactivities (antimicrobial, antioxidant, and cytotoxic) were studied. [2] The compounds responsible for the bioactivity were not elucidated. [3] Information not available.

*2.2. Biosynthetic Pathways and Structure of Streptomyces Pigments*

Among the *Streptomyces* pigments are prodiginins, such as prodigiosin (4), undecyl-prodigiosin (2), streptorubin B (3) and metacycloprodigiosin (1) (Figure 6); derivatives of naphthoquinones (5–7); actinomycins, such as actinomycins $X_2$ (10), actinomycins $L_1$ (8), and actinomycins $L_2$ (9); actinorhodins, such as γ-Actinorhodin (11), λ-Actinorhodin (12), and actinorhodin (13); grixazones such as grixazone A (14) and B (15); melanin, including eumelanin and pyomelanin; other compounds, such as indigoidine (16), katorazone (17) and 4,8,13-trihydroxy-6,11-dione-trihydrogranaticins A (TDTA) (18). However, despite the large number of studies, some structural aspects and their biosynthetic pathways require further study.

**Figure 6.** Chemical structures of prodiginins.

Prodiginine pigments are characterized by a common pyrrolyl dipyrromethene skeleton. Especially, bacterial prodiginines have been divided into linear, which include prodigiosin (4) and undecylprodigiosin (2), and cyclic derivatives, which include streptorubin B (3) and metacycloprodigiosin (1). Biosynthesis of the prodiginines proceeds via a bifurcated pathway, culminating in the enzymic condensation of the bipyrrole, 4-methoxy-2-2′-bipyrrole-5-carbaldehyde (MBC) with either 2-methyl-3-pentylpyrrole (MPP) to form prodigiosin (4) [111].

Quinones are aromatic compounds widely present in nature (Figure 7). They can be classified according to their chemical structures into benzoquinones, anthraquinones and naphthoquinones. Specifically, naphthoquinones are structurally related to naphthalene and are characterized by their two carbonyl groups in the 1,4 position or 1,2 position with minor incidence; they are highly reactive organic compounds used as dyes whose colors range from yellow to red [112].

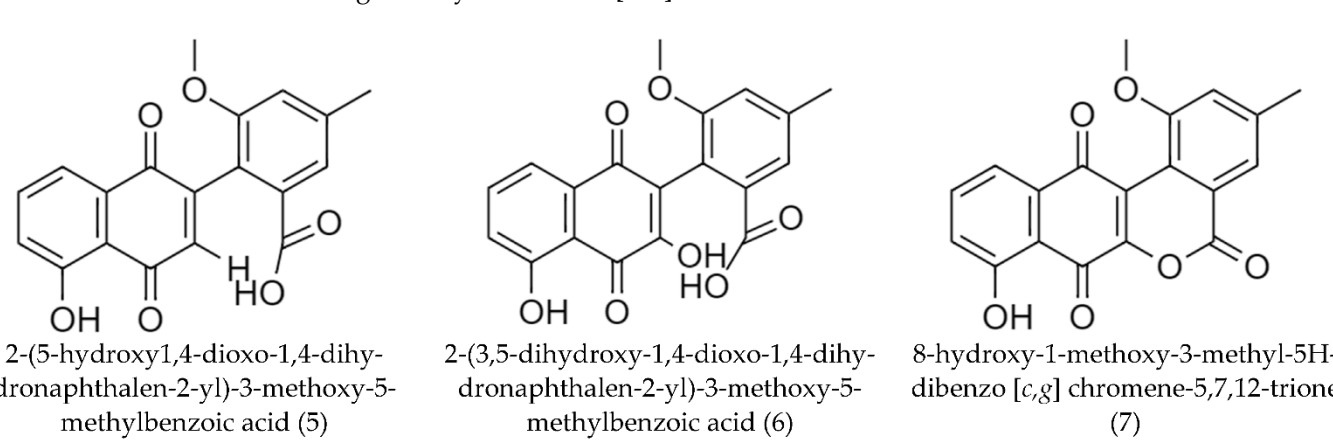

2-(5-hydroxy1,4-dioxo-1,4-dihy-dronaphthalen-2-yl)-3-methoxy-5-methylbenzoic acid (5)　　　2-(3,5-dihydroxy-1,4-dioxo-1,4-dihy-dronaphthalen-2-yl)-3-methoxy-5-methylbenzoic acid (6)　　　8-hydroxy-1-methoxy-3-methyl-5H-dibenzo [*c,g*] chromene-5,7,12-trione (7)

**Figure 7.** Chemical structures of naphthoquinones.

Actinomycin is a DNA-targeting antibiotic and anticancer, composed of a chromophore group and two pentapeptide chains with a variable composition of amino acids. The pentapeptide precursors are biosynthesized by a nonribosomal peptide synthetase (NRPS) assembly line, and actinomycins are formed through oxidative condensation of two 3-hydroxy-4-methylanthranilic acid (4-MHA) pentapeptide lactones (PPLs) [80].

Actinomycins $X_2$ (10) are formed through the sequential oxidation of the γ-prolyl carbon by the cytochrome P450 enzyme saAcmM. Actinomycin L (8,9) is formed through the spontaneous reaction of anthranilamide with the 4-oxoproline site of actinomycin $X_2$ (10) prior to the condensation of the two 4-MHA PPLs into actinomycin L (8,9) [80] (Figure 8).

Actinorhodin (13) is a blue pigment whose polyketide backbone must undergo two regiospecific reductions, two intramolecular aldol condensations, hemiketalization, aromatization of two rings, oxidation to form a quinone, hydroxylation, and dimerization [113] (Figure 9). This pigment is a redox-active secondary metabolite and a potent, bacteriostatic, pH-responsive antibiotic. Additionally, it is redox-active and can act in redox-cycling reactions. Moreover, it act as an organocatalyst of oxidative reactions in vitro, which suggests that actinorhodin (13) might kill bacteria via the accumulation of toxic concentrations of $H_2O_2$ [114].

Actinomycin L₁ (8)

Actinomycin L₂ (9)

Actinomycin X₂ (10)

**Figure 8.** Chemical structures of actinomycins.

γ-Actinorhodin (11)

λ-Actinorhodin (12)

Actinorhodin (13)

**Figure 9.** Chemical structures of actinorhodins.

The act PKS includes the minimal PKS components (KS, CLF, and ACP), which together synthesize the octaketide backbone, a C-9 ketoreductase (KR), a didomain aromatase/cyclase (ARO/CYC) which is required for the formation of the first aromatic ring, and a second ring cyclase (CYC2). Additionally, based on the above PKS definition, the C-3 enoyl reductase and the third ring cyclase/dehydratase (if one exists) may also associate with the PKS complex [113].

Grixazone contains a phenoxazinone chromophore. Especially, grixazone A(14) is a novel compound, and grixazone B (15) has been reported to show a parasiticide activity [115] (Figure 10); expression of the biosynthetic genes for this yellow pigment is probably under the control of A-factor (factor (2-isocapryloyl-3R-hydroxymethyl-γ-butyrolactone)), which triggers the synthesis of almost all of the secondary metabolites produced by *Streptomyces griseus* [115], and controlled by the phosphate concentration of the medium [98].

Grixazone A (14)                                          Grixazone B (15)

**Figure 10.** Chemical structures of grixazones.

In the grixazone biosynthesis gene cluster, griF (encoding a tyrosinase homolog) and griE (encoding a protein similar to copper chaperons for tyrosinases) are encoded. GriF is thus a novel o-aminophenol oxidase that is responsible for the formation of the phenoxazinone chromophore in the grixazone biosynthetic pathway. No study on the precursor(s) or the biosynthetic enzyme for the phenoxazinone skeleton of these compounds has so far been reported. Because grixazones A (14) and B (15) contain an aldehyde and a carboxyl group, respectively, at the 8-position, the biosynthesis of the phenoxazinone skeleton in grixazones should be the same as those in michigazone and texazone [115].

Indigoidine (16) is a member of the class of pyridone, an extracellular blue pigment from *S. aureofaciens* CCM 3239 (Figure 11). The bpsA gene, which encodes nonribosomal peptide synthetase, is responsible for the biosynthesis of the blue pigment indigoidine (16) [116]. Novakoba et al. [95] determined that a deletion mutant of bpsA in *S. aureofaciens* CCM 3239 failed to produce the blue pigment and had a positive effect on auricin production, indicating the involvement of the bpsA gene in the biosynthesis of the indigoidine (16) blue pigment in *S. aureofaciens* CCM 3239 [95].

Indigoidine (16)

**Figure 11.** Chemical structures of indigoidine.

Katorazone (17) is an alkaloid with a 2-azaquinone-phenylhydrazone structure (Figure 12). Characteristic structural features of 2-azaquinones in nature are the presence of a methyl group at C-3 and different substituents in ring C [87]. The biosynthetic pathway of katorazone (17) of bacterial origin is unclear; however, it is a derivative of anthraquinones, which in plants have two main biosynthetic pathways: the polyketide pathway and the chorismate/o-succinylbenzoic acid pathway [117].

Katorazone (17)

**Figure 12.** Chemical structures of katorazone.

4,8,13-trihydroxy-6,11-dione-trihydrogranaticins A (TDTA) (18) is a type of granaticin, which is a benzoisochromanequinone that is structurally very similar to actinorhodin (13) (Figure 13); however, the stereochemistry around the oxygen bridge of the pyran ring in granaticin is opposite to that of actinorhodin (13) and the presumed tricyclic intermediate undergoes *C*-glycosylation instead of dimerization. The *gra* gene cluster encodes for the biosynthesis and transfer of the appropriate deoxysugar group. The *gra* PKS genes include the minimal PKS genes, a KR gene, and an ARO/CYC gene. Further sequencing of this gene cluster has led to the identification of several genes involved in deoxysugar biosynthesis [113].

TDTA (18)

**Figure 13.** Chemical structures of TDTA.

Resistomycin (19) is a pentacyclic polyketide metabolite and quinone-related antibiotic, which has a unique structure—a ring system that differs from other bacterial aromatic polyketides [93] (Figure 14). Jakobi et al. [118] identified the entire gene cluster encoding resistomycin (19) biosynthesis and determined that the *rem* gene cluster exhibits several unusual features of the type II PKS involved, most remarkably a putative malonyl CoA acyltransferase (MCAT) with highest homology to AT domains from modular PKSs [118].

**Figure 14.** Chemical structures of resistomycin.

Otherwise, melanins (Figure 15) are polymers with diverse structures and brown to black colorations [116]. They have multiple important functions and are formed by the oxidative polymerization of phenol and/or indolic compounds; however, their structures are not well understood [55]. Actinobacteria members produce a dark pigment, melanin, which is considered valuable for taxonomic relatedness [116].

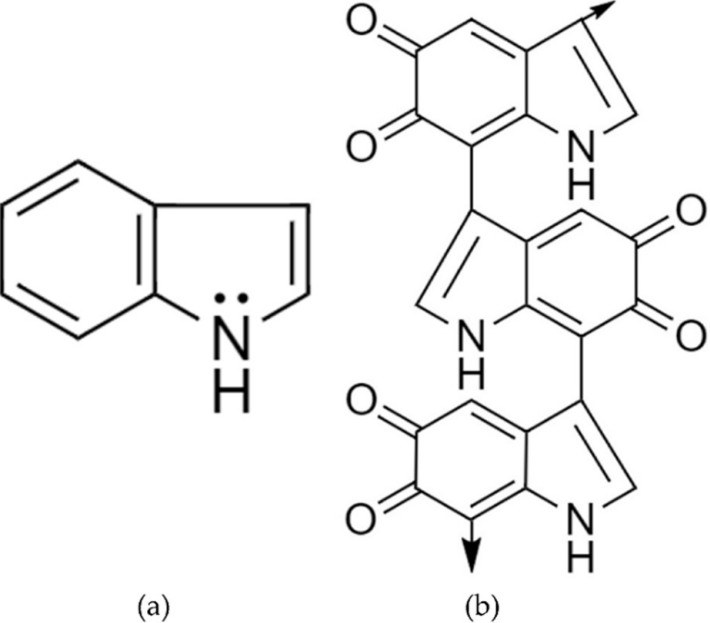

(a)                                        (b)

**Figure 15.** Melanin-related structures and some properties. (**a**) Basic structure (indolic ring). (**b**) Resonance structures that are probably involved in the process of color. The arrows show the points and sense of polymerization [119].

The enzyme tyrosinase is responsible for the first step in the melanin biosynthesis and the gene *melC* encodes for the tyrosinase operon [116]. The regulatory region of the *mel* gene has three unique sites for the *Sst*I, *Bgl*II, and *Sph*I restriction endonucleases, which permits the easy recognition of colonies containing the insertion of the DNA of interest [116].

There are five types of melanins (eumelanin, pyomelanin, pheomelanin, allomelanin, and neuromelanin), and each of these pigments is synthesized enzymatically or nonenzymatically from different precursors by different metabolic pathways [107].

Eumelanin is a polymer of 5,6-dihydroxyindole (DHI) and 5,6-dihydroxyindole-2- carboxylic acid (DHICA) [55], originating from the tyrosine or phenylalanine amino acids [107]; however, the detailed polymer structure is undetermined [55]. The pathway of eumelanogenesis may be divided into two phases, one proximal and the other distal. The proximal phase consists of the enzymatic oxidation of tyrosine or L-DOPA to its corresponding

dopaquinone catalyzed by tyrosinase and the distal phase is represented by chemical and enzymatic reactions which occur after dopachrome formation and lead to the synthesis of eumelanins [55].

Pyomelanin is a natural polymer of homogentisic acid (HGA, 2,5-dihydroxyphenylacetic acid) synthesized through the L-tyrosine pathway, and belongs to the heterogeneous group of allomelanins [120]. It is formed by the catabolism of tyrosine and/or phenylalanine [90].

### 2.3. Pigment Purification

The initial process, in most cases, wherein the pigment is secreted into the culture broth, is centrifugation to remove the biomass [66,85]. To partially separate the pigment from the other metabolites that generally accompany it, different types of reagents are applied, such as acids or bases. To modify the pH and make the target pigment precipitate [40,66], solid–liquid extraction is used [94,102]. To add solvents in different ratios (organic, polar, and nonpolar) [63] or to establish a solvent system to extract the pigment [84], liquid–liquid extraction is used [79]. In addition, different conditions are used that modify the pressure or the temperature to concentrate the pigment, such as vacuum-freeze-drying or lyophilization [40,66,67], flash evaporation [60,108] or reduction of pressure [61], and rotary evaporation [79,99]. Some research has used physical methods to assist in the purification of intracellular pigments, such as using an ultrasonic cell-disrupter system to perform lysis (sonicated) [82,105], filtration using membranes [98] and ultrafiltration membranes [100], commercial products such as a centricon-30 concentrator [100], and a Soxhlet extractor [106]. The number of times a specific procedure is repeated on the purification strategy is variable. For example, performing centrifugation two or three times [60,103,104,107], performing a re-extraction many times [61,82], and repeated crystallizations [106].

Once a crude extract containing the pigment is available, different separation techniques are applied, among which the most used is chromatography. Depending on the physicochemical characteristics of the pigment, the most appropriate one will be chosen.

Among the types of chromatography used for pigment purification are flash chromatography [88], column chromatography on alumina [64], thin-layer chromatography (TLC) [46,55,59], ion-exchange chromatography [66], high-performance liquid chromatography (HPLC) [71,92], and preferentially coupled to mass spectroscopy (MS) [77,86,93]. Thus, between some of the columns used are SephadexG-50 [66,67], silica gel columns [61,85], Kromasil ODS C-18 columns [86], Sephadex LH-20 [87,91,103,104], Zorbax ODS C18 [97], Silica-coated glass plates [97], silica-gel-coated aluminum sheets [99], and DEAE-Sepharose columns [100]. In special cases, the compounds separated were detected by UV illuminator [74].

Additionally, it should be clarified that some of the purification processes allow one to concentrate the pigment. These include lyophilization or freeze-drying, which is a dehydration process based on the sublimation of ice contained in the material, the above in order to obtain a final product with little damage caused by thermal and chemical degradation [121].

Yield is one of the major concerns when scaling up a process. In this case, the pigments produced by *Streptomyces* are very variable but most of them are high or can be optimized to achieve better yields. Table 2 shows some of the selected articles that describe pigment purification. The yields, based on the dry weight of the pigment from the initial volume of the fermentation broth, are reported (see Table 2). The highest yields for melanin is from *Streptomyces* sp. (strains F1, F2, and F3) [60]; red pigment is from *Streptomyces* sp. PM4 [61], yellow pigment is from *Streptomyces griseoaurantiacus* JUACT 01 [84], and blue pigment is from *Streptomyces coelescens* ATCC 19830 (NP2) [96].

**Table 2.** Yield reported for each type of pigment and the *Streptomyces* strain that produces it.

| *Streptomyces* Strain | Type/Color of Pigment | Yield Reported (mg/L) | Ref. |
|---|---|---|---|
| *Streptomyces* sp. MVCS13 | Melanin | 239 | [66] |
| *Streptomyces* sp. F1 <br> *Streptomyces* sp. F2 <br> *Streptomyces* sp. F3 | Melanin | 21130 | [60] |
| *Streptomyces* sp. | Melanin | 1460 | [67] |
| *Streptomyces glaucescens* NEAE-H | Melanin | 350 | [40] |
| *Streptomyces* sp. ZL-24 | Melanin | 59 to 138 | [94] |
| *Streptomyces glaucescens* KCTC988 | Melanin | 125 [1] | [102] |
| *Streptomyces cavourensis* SV 21 | Melanin | 670 | [70] |
| *Streptomyces parvus* BSB49 | Eumelanin | 160 to 240 | [90] |
| *Streptomyces sp.* CWW6 | Streptorubrin A (prodiginine pigment) | 20 | [64] |
| *Streptomyces* sp. WMA-LM31 | Prodigiosin (4) | 30 | [91] |
| *Streptomyces* sp. JS520 | Red pigment | <1 to 139 [1] | [88] |
| *Streptomyces* sp. PM4 | Red pigment | 1874 | [61] |
| *Streptomyces acidiscabies* | Naphthoquinone derivatives: <br> Bright yellow compound (5) [2] <br> Orange compound (6) [3] <br> Compound (7) [4] | 63 <br> 28 <br> 2 | [72] |
| *Streptomyces* sp. D25 | Yellow pigment | 175 to 1225 [5] | [62] |
| *Streptomyces griseoaurantiacus* JUACT 01 | Yellow pigment | 4300 | [84] |
| *Streptomyces aurantiacus* AAA5 | Resistomycin (19) (yellow compound) | 52 | [93] |
| *S. griseus* IFO13350 w | Grixazone A (14) <br> Grixazone B (14) | 5 <br> 2 | [98] |
| *Streptomyces* sp. A1013Y | Blue pigment | 2 | [82] |
| *S. coelicolor* 100 | Blue pigment | 3000 | [86] |
| *Streptomyces coelicolor* MSIS1 | Red or blue, depending on conditions | 5030 (shake flasks) <br> 9000 (bioreactor) | [92] |
| *Streptomyces coelescens* ATCC 19830 *(NP2)* | Deep blue | 3000 to 4000 | [96] |
| *Streptomyces anthocyanicus* ATCC 19821 *(NP4)* | Deep red | 3000 to 4000 | [96] |

[1] In the original article, these results were presented with the standard deviation. [2] The compound name is 2-(5-hydroxy-1,4-dioxo-1,4-dihydronaphthalen-2-yl)-3-methoxy-5-methylbenzoic acid. [3] The compound name is 2-(3,5-dihydroxy-1,4-dioxo-1,4-dihydronaphthalen-2-yl)-3-methoxy-5-methylbenzoic acid. [4] The compound name is 8-hydroxy-1-methoxy-3-methyl-5H-dibenzo [*c,g*] chromene-5,7,12-trione. [5] It reported the effect of critical medium components and culture conditions in mg/40 mL, so the unit's conversion is made, and a range is presented.

Additionally, the largest scale-up was from 800 mL (8 Erlenmeyer of 250 mL with 100 mL of medium) to a 5 L bioreactor with 3 L of medium; this scaling up by increasing the agitation from 200 to 300 rpm and adding an air flow of 3 L/min resulted in a significant increase in pigment production yields (from 5030 to 9000 mg/L) [92].

### 2.4. Stability Tests of the Pigments

Even though one of the main disadvantages of pigments is their stability, only 17% of the selected articles evaluated them. The stability of the pigment includes assays under

different conditions of pH, temperature, or light [76,97], or, in the presence of metal ions, additives, vitamins, and reducing and oxidizing agents [82,86].

Pigment stability against a wide pH range was the most evaluated, followed by photo-stability (UV light, outdoor sunshine, and indoor incandescent light and dark) [76,82] and thermostability in equal measure. Next, the most evaluated assay in the same proportion were stability in the presence of metal ions ($Fe^{3+}$, $Fe^{2+}$, $Pb^{2+}$, $Mg^{2+}$, $Cu^{2+}$, $Al^{3+}$, $Ca^{2+}$, $Zn^{2+}$, and $Na^+$) and stability in presence of additives (citric acid, malic acid, sodium hydroxide, sodium citrate, sodium benzoate, glucose, sucrose, maltose) and/or vitamins (VA, VD3, VE, VB1, VB2, VB3, VB5, VB6, and VB9) [82,86]. Finally, the least evaluated assay is stability in presence of reducing and oxidizing agents [86].

Most of the *Streptomyces* pigments reported are very stable (see Table 3), which is a relevant fact for its potential applications; however, the condition at which some are most unstable is pH. TDTA (18) pigments from *Streptomyces* sp. A1013Y and λ-actinorhodin (12) from *Streptomyces coelicolor* 100, especially, were subjected to multiple stability tests and the only disadvantage is their sensitivity to different pHs [82,86].

**Table 3.** Stability reported for each type of pigment and the *Streptomyces* strain that produces it.

| *Streptomyces* Strain | Type/Color of Pigment | Stability Results | Ref. |
|---|---|---|---|
| *Streptomyces vietnamensis* sp. nov. GIMV4.0001 | Violet–blue pigment | pH-sensitive<br>Stable at high temperature<br>Stable under UV light | [76] |
| *Streptomyces spectabilis* L20190601 | Metacycloprodigiosin (1) | pH-sensitive, red at pH 3.0 and yellow or orange at pH 9.0 | [77] |
| *Streptomyces* sp. A1013Y | TDTA (18) | Stable in a wide range of pHs<br>Thermo-stable<br>Good stability with indoor incandescent and ultraviolet lights, but unstable with sunlight<br>Stable with most metal ions and vitamins except $Fe^{3+}$, $Cu^{2+}$, and $Al^{3+}$ | [82] |
| *Streptomyces coelicolor* 100 | λ-Actinorhodin (12) | pH-sensitive; red at pH < 7, amaranth at pH 7–8, blue at pH > 8<br>Photo-stable<br>Thermo-stable<br>Resistant to oxidants and reducers under acid conditions and to reducers under alkaline conditions.<br>Stable with food additives<br>Stable with most metal ions except $Fe^{2+}$ and $Pb^{2+}$ | [86] |
| *Streptomyces coelicolor* MSIS1 | May be one of actinorhodinic acid. | pH-sensitive, red at pH < 7, amaranth at pH 7–8, blue at pH > 8 | [92] |
| *Streptomyces coelescens* ATCC 19830 (NP2) | Deep blue | pH-sensitive | [96] |
| *Streptomyces anthocyanicus* ATCC 19821 (NP4) | Deep red | pH-sensitive | [96] |
| *Streptomyces* sp. LS-1 | May be actinorhodin-related compounds. | Sensitive to low pH<br>Relatively photo-stable<br>Relatively thermo-stable | [97] |
| *Streptomyces parvullus* M4 | Red | pH-Stable | [105] |
| *Streptomyces coelicolor* M6 | Red | pH-Stable | [105] |
| *Streptomyces cyaneofuscatus* | Actinomycin X2 (10) | Excellent thermal stability<br>Acid and alkali resistance | [69] |

### 2.5. Optimization of the Pigment's Production

One of the main concerns in pigment production is yield. For this reason, the optimization of culture media and fermentation conditions becomes important. There are different ways to improve the amount of the pigment: by changing the kind of medium used, including the ISP (International *Streptomyces* Project) medium, among others; changing the concentration of the components of a base culture medium; changing the main sources of nutrients, such as carbon and nitrogen sources (see Table 4); even using experimental designs such as Plackett–Burman and central composite design with multiple factors and levels [43,73,78].

The main most optimized variables are the sources of carbon and nitrogen and their concentrations: carbon sources include glycerol [66], starch, dextrose, maltose, fructose, glucose [60,61], xylose, arabinose, rhamnose, galactose, raffinose, mannitol, inositol, sucrose [71], lactose [73], melibiose, and sorbinose [76]. Nitrogen sources include L-tyrosine or tyrosine, asparagine [61,66], yeast extract, soyabean meal, peptone [60,61], ammonium sulphate, malt extract, phenyl alanine, histidine [61], glutamine [62], potassium nitrate, protease-peptone, ferric ammonium citrate [40], sodium nitrate ($NaNO_3$), ammonium chloride ($NH_4Cl$), beef extract, casein, casein peptone, meat peptone [89], soy peptone [94], $KNO_3$ [97], ammonium nitrate [103], and urea [108].

Taking into account the economy and the large amount of agroindustrial waste, there are new sources of nutrients being used, such as sugar cane waste, rice bran, wheat bran, coconut cake, and rice flour [60]. Other components of culture media that have been used to optimize pigment production include salts, such as $MgSO_4$, NaCl, $FeSO_4$, $K_2HPO_4$ [66], $Na_2HPO_4$ [61], $CaCO_3$, KCl [62], sodium thiosulfate [40], $KH_2PO_4$, $CaCl_2$, $NiCl_2$ [94], $FeCl_2$, $MgCl_2$, $ZnCl_2$, $MnCl_2$,$CaCl_2$, and $CoCl_2$ [108]; and amino acids, such as glycine, cystine, alanine, tryptophan, valine [71], tryptone [73], leucine, proline, glutamine [103], aspartate, and glutamate [108].

Other variables that have been considered in optimizing pigment production are Ph (5–9.5), temperature (10–60 °C) [60,66,73], salinity (0–20 ppt) [60], concentration of NaCl (1%–10%) [44,53,89], incubation period (3–15 days) [89], agitation speed (50–200 rpm) [88], medium volume (mL/250 mL flask) [40], inoculation size [94], and sources and concentration of phosphate [103].

**Table 4.** Optimization conditions of pigments produced by each *Streptomyces* strain.

| *Streptomyces* Strain | Type/Color of Pigment | Yield Reported (mg/L) | Optimized Variable | Optimization Result | Ref. |
|---|---|---|---|---|---|
| *Streptomyces* sp. MVCS13 | Melanin | 239 | Temperature<br>pH<br>L-Tyrosine<br>Asparagine<br>$MgSO_4$<br>NaCl<br>$FeSO_4$<br>Trace salt solution | 50 °C<br>7.4<br>0.75 g/L<br>1.5 g/L<br>0.25 g/L<br>0.75 g/L<br>0.015 g/L<br>1.5 mL/L | [66] |
| *Streptomyces* sp. F1<br>*Streptomyces* sp. F2<br>*Streptomyces* sp. F3 | Melanin | 21,130 | Carbon Source<br>Nitrogen source<br>Salinity<br>Temperature<br>pH<br>Incubation time<br>Cheaper source | Starch 1% *w/v*<br>Soyabean 0.2% *w/v*<br>15 ppt<br>35 °C<br>7<br>168 h<br>Sugarcane waste | [60] |
| *Streptomyces* sp. PM4 | Red pigment | 1874 | Carbon Source<br><br>Nitrogen source | Maltose (4.06 g/L)<br>Peptone (7.34 g/L)<br>Yeast extract (4.34 g/L)<br>Tyrosine (2.89 g/L) | [61] |

**Table 4.** *Cont.*

| *Streptomyces* Strain | Type/Color of Pigment | Yield Reported (mg/L) | Optimized Variable | Optimization Result | Ref. |
|---|---|---|---|---|---|
| *Streptomyces* sp. AQBWWS1 | Carotenoid | N/A [1] | Carbon Source<br>Amino acids<br>NaCl Concentration | Glucose<br>Xylose<br>Cystine<br>Tryptophan<br>2.50% | [71] |
| *Streptomyces* sp. D25 | Yellow pigment | 1225 | Carbon Source<br>Nitrogen source<br>pH<br>Temperature<br>NaCl Concentration | Glucose<br>Fructose<br>Malt Extract<br>7, 9, 11<br>30 °C, 40 °C<br>1%–5% | [62] |
| *Streptomyces* sp. S45 | Pinkish-brown pigment | N/A | Carbon Source<br>Nitrogen source<br>Minerals<br>pH<br>Temperature | Glucose<br>Rhamnose<br>Soybean meal<br>CaCl$_2$<br>7<br>30 °C | [63] |
| *Streptomyces glaucescens* NEAE-H | Melanin | 350 | Incubation period<br>Nitrogen source | 6 days<br>Protease-peptone (5 g/L)<br>Ferric ammonium citrate (0.5 g/L) | [40] |
| *Streptomyces* sp. ZL-24 | Melanin | 138 | NiCl$_2$<br>FeSO$_4$<br>Soy peptone<br>pH<br>Temperature<br>Inoculation size<br>Incubation period | 3.05 Mm<br>1.33 g/L<br>20.31 g/L<br>7<br>30 °C<br>3% (*v/v*)<br>5 days | [94] |
| *Streptomyces* sp. LS-1 | May be actinorhodin-related compounds. | N/A [1] | Carbon Source<br>Nitrogen source | Glucose<br>KNO$_3$ | [97] |
| *Streptomyces canaries* M8 | Carotenoid | N/A | NaCl Concentration | >10% | [105] |
| *Streptomyces* sp. Ac-1 | Yellow pigment | N/A [2] | Agitation<br>NaCl Concentration<br>pH | 100 rpm<br>2%<br>5 | [65] |
| *Streptomyces* sp. Ac-2 | Yellow pigment | N/A [2] | Agitation<br>NaCl Concentration<br>pH | Steady state<br>4%<br>9 | [65] |
| *Streptomyces* sp. | Red pigment | N/A | Temperature<br>pH (Solid media)<br>pH (Broth culture) | 37 °C<br>10.5 or 7<br>7 | [68] |

[1] The result of the optimization was expressed as absorbance of the supernatant at 590 nm. [2] The result of the optimization was expressed as dry weight of the biomass (g).

*2.6. Bioactivity Results*

2.6.1. Antimicrobial Activity

Antibiotics are essential for human health and are one of the pillars of modern medicine; however, we are dealing with the evolution and dissemination of resistance mechanisms that endangers the current arsenal of antibiotics. One of the requirements in this matter is the discovery of new antimicrobial compounds with high efficiency and nontoxicity [80,122,123]. In this way, actinobacteria and especially the *Streptomyces* genus are responsible for most antibiotics in use today [48,80,124].

The principal methodology to evaluate the antimicrobial activity is the disk-diffusion method. Different microorganisms are used to evaluate the antimicrobial activity: Gram-negative bacteria, such as *E. coli* [63,76–78], *Salmonella* sp. [60,74], *K. pneumoniae* [40,89,92,93], and *P. aeruginosa* [93,94] (see Table S2)*;* Gram-positive bacteria, such as VRSA (vancomycin-resistant *Staphylococcus aureus*), MRSA (methicillin-resistant *Staphylococcus aureus*) [75], *Enterococcus* sp. [74,88], and *Nocardia asteroides* [64] (see Table S3); fungi, such as *Aspergillus niger* [40] and *Fusarium oxysporum* [94]; and yeast, such as *Saccharomyces cerevisiae* [98] (see Table S4). Studies have even begun to evaluate the bioactivity of some strains, such as *Streptomyces* sp. S45, and how they act against human immunodeficiency virus, showing an $IC_{50}$ value of 8.75 µg/mL [63].

Many of the *Streptomyces* strains that produce pigments have antimicrobial activity reporting the measure of the inhibition zone (see Table 5) and, in other cases, were determined to have minimal inhibitory concentrations (MIC) (see Table 6). Thus, the reported *Streptomyces* strains are promising antibiotic producers against innumerable pathogens.

**Table 5.** *Streptomyces* strains with antimicrobial activity.

| *Streptomyces* Strains | Source | Positive Antimicrobial Tests | Inhibition Zone (mm) | Ref. |
|---|---|---|---|---|
| *Streptomyces* sp. F1 *Streptomyces* sp. F2 *Streptomyces* sp. F3 | Pure colonies | *E. coli, Lactobacillus vulgaris, Proteus mirabilus, Vibrio cholera, S. aureus, S. typhi, S. paratyphi,* and *K. oxytoca* | N/A | [60] |
| *Streptomyces coeruleorubidus* NBRC 12844 | Extract | *S. aureus* ATCC 1112, *B. cereus* ATCC1015, *P. aeruginosa* ATCC 1074, *C. freundii* ATCC 8090, *K. pneumoniae* ATCC 1053, and *S. marcescens* ATCC 14756 | N/A | [73] |
| *Streptomyces* sp. D10 | Fraction | *MRSA* *VRSA* *E. coli* *Klebsiella* sp. | 15 20 15 10 | [75] |
| *Streptomyces* sp. D25 | Extract | *MRSA* | 22 [1] | [62] |
| *Streptomyces* sp. SAG-85 | Extract | *MRSA* *S. marcescens* | 23 [2] 47 [2] | [78] |
| *Streptomyces* sp. ZL-24 | Compound [3] (Melanin) | *P. aeruginosa* ATCC 9027 *E. coli* ATCC 8379 *S. aureus* ATCC6538 *Mycobacterium smegmatis* ATCC 10231 | 26–32 21–29 15–30 17–36 | [94] |
| *Streptomyces coelicolor* A3(2) | Extract | *B. subtilis, S. aureus, E. coli,* and *Pseudomonas fluorescens* | N/A | [101] |
| *Streptomyces cyaneofuscatus* | Compound (Actinomycin X2 (10)) | *S. aureus* ATCC 6538 | 20 | [69] |
| *Streptomyces* sp. Ac-2 | Extract | *P. aeruginosa* ATCC 27853 *S. aureus* ATCC 25923 *E. coli* ATCC 25922 | 18 [2] 22 [2] 21 [2] | [65] |

**Table 5.** *Cont.*

| *Streptomyces* Strains | Source | Positive Antimicrobial Tests | Inhibition Zone (mm) | Ref. |
|---|---|---|---|---|
| *Streptomyces* sp. | Extract [4] | *S. aureus* MTCC 3160<br>*B. subtilis* MTCC 736<br>*E. coli* MTCC 1554<br>*Vibrio cholera* MTCC 3906 | 4<br>10<br>6<br>5 | [68] |
| *Streptomyces* sp. NS-05 | Extract [5] | *E. coli* MTCC 739<br>*Proteus vulgaris* MTCC 6380 | 5 [2]<br>8 [2] | [81] |

[1] This is the average of the extracts with activity in different solvents (methanol, dichloromethane, diethyl ether extract, chloroform extract, and ethyl acetate). [2] The original data were reported with standard deviations. [3] This is a range; the two forms of melanin (insoluble and soluble) were evaluated at different concentrations. [4] The measure presented is the maximum inhibition zone of different concentrations. [5] Additionally, there is information about antimicrobial activity of biopigment-assisted synthesized nanoparticles.

**Table 6.** *Streptomyces* strains with antimicrobial activity and its MIC values.

| *Streptomyces* Strain (Source/Compound Name) | Positive Antimicrobial Tests | MIC (µg/mL) | Ref. |
|---|---|---|---|
| *S. spectabilis* L20190601 [1] (Metacycloprodigiosin (1)) | *Staphylococcus aureus*<br>*Bacillus subtilis*<br>*Escherichia coli*<br>*Streptococcus pyogenes*<br>*Pseudomonas aeruginosa*<br>*Bacillus typhi*<br>*Candida albicans*<br>*Trichophyton rubrum* | <1<br><1<br>4<br><1<br><1<br>1<br>2<br>64 | [77] |
| *Streptomyces* sp. JS520 [1] (Undecylprodigiosin (2)) | *Micrococcus luteus* ATCC 379<br>*Bacillus subtilis* ATCC 6633<br>*Candida albicans* ATCC 10231<br>*Candida albicans* ATCC 10259 | 50<br>50<br>100<br>200 | [88] |
| *Streptomyces* sp. JAR6 [1] (Undecylprodigiosin (2)) | *Salmonella* sp.<br>*Bacillus subtilis*<br>*Proteus mirabilis*<br>*Shigella* sp.<br>*Escherichia coli*<br>*Enterococcus* sp.<br>*Klebsiella pneumoniae* | 150<br>50<br>80<br>100<br>170<br>120<br>180 | [89] |
| *Streptomyces* sp. MVCS13 (Melanin) | *Bacillus* sp. FPO1<br>*Aeromonas* sp. FPO2<br>*Citrobacter* sp. FPO3<br>*Edwardsiella* sp. FPO4<br>*Vibrio* sp. FPO5<br>*Aeromonas* sp. FPO6 | 23 [2]<br>27 [2]<br>21 [2]<br>20 [2]<br>18 [2]<br>22 [2] | [66] |
| *Streptomyces aurantiacus* AAA5 (Resistomycin (19)) | *S. epidermis*<br>*Enterococcus faecalis*<br>*Bacillus subtilis*<br>*Staphylococcus aureus*<br>*Klebsiella pneumoniae*<br>*Shigella* sp.<br>*Proteus vulgaris*<br>*Escherichia coli*<br>*Pseudomonas aeruginosa*<br>*Salmonella typhii* | 8 [2]<br>5 [2]<br>25 [2]<br>13 [2]<br>16 [2]<br>45 [2]<br>70 [2]<br>42 [2]<br>34 [2]<br>15 [2] | [93] |

**Table 6.** *Cont.*

| *Streptomyces* Strain (Source/Compound Name) | Positive Antimicrobial Tests | MIC (µg/mL) | Ref. |
|---|---|---|---|
| *Streptomyces* sp. MBT27 [1] (Actinomycins $L_1$ (8)) [3] | *Staphylococcus aureus* MB5393 | 4–8 | [80] |
| | *Staphylococcus aureus* ATCC29213 | 2–4 | |
| | *Vancomycin-sensitive Enterococcus faecium* | 4–8 | |
| | *Vancomycin-resistant Enterococcus faecium* | 4–8 | |
| | *S. epidermidis* | 4–8 | |
| | *Escherichia coli* ATCC25922 | >128 | |
| | *Klebsiella pneumoniae* ATCC700603 | >128 | |
| *Streptomyces* sp. MBT27 [1] (Actinomycins $L_2$ (9)) [3] | *Staphylococcus aureus* MB5393 | 8–16 | [80] |
| | *Vancomycin-sensitive Enterococcus faecium* | 8–16 | |
| | *Vancomycin-resistant Enterococcus faecium* | 8–16 | |
| *Streptomyces parvulus* C5-5Y (Fraction F5) | *S. aureus* | 125 | [74] |
| | *S. epidermidis* | 125 | |
| | *Enterococcus faecalis* | 250 | |
| | *E. coli* | 375 | |
| | *Pseudomonas* sp. | 125 | |
| | *K. pneumoniae* | 125 | |
| | *S. typhi* | 125 | |
| | *Proteus vulgaris* | 500 | |
| | *Shigella* sp. | 125 | |
| | *Streptococcus mutans* | 125 | |
| *Streptomyces* sp. S45 (Fraction) | *S. aureus* ATCC 29213 | 2 | [63] |
| *Streptomyces* sp. BSE6.1 (Prodigiosin (4)) | *S. aureus* MTCC1430 | 400 | [99] |

[1] Only performed the antimicrobial assay once and did not report their respective standard deviations. [2] The original data were reported with standard deviations. [3] These compounds are pigment-associated compounds; however, its role as a pigment or its coloration are not clear.

Especially, *Streptomyces* sp. D25 was evaluated against *M. tuberculosis H37Rv* using luciferase reporter mycobacteriophage (LRP) assays, and the results were expressed in the percentage reduction in the relative light unit (RLU). In this case, they evaluated extracts in different solvents (methanol, dichloromethane, diethyl ether extract, chloroform extract, and ethyl acetate) and the activity ranged from 84.74 ± 3.60 to 91.59 ± 4.02 [62].

The minimum inhibitory concentration varies widely (0.5–200 µg/mL). Especially, the *S. spectabilis* strain L20190601 and *Streptomyces* sp. MBT27 have antimicrobial activity, even at very low concentrations [77] (see Table 6). Of the few that reported an MIC value, only some performed the assay more than once and reported their respective standard deviations.

The prodiginine family are primarily red-pigmented, specialized metabolites that have a tripyrrole structure [125] and include compounds such as undecylprodigiosin (2) and metacycloprodigiosin (1). These compounds are promising antimicrobials against Gram-positive and Gram-negative bacteria and fungi; therefore, this family can be an inexhaustible source of antibiotics [77,88,89]. On the other hand, pigments such as the melanin from *Streptomyces* sp. MVCS13 have a potential effect against the ornamental fish pathogens of *Carassius auratus* [66]. Additionally, even though actinomycin $L_1$ (8) and $L_2$ (9) from *Streptomyces* sp. MBT27 are diastereomers that stem from the aminal formation at C-10′, actinomycin $L_1$ (8) showed a somewhat higher bioactivity than actinomycin $L_2$ [80].

2.6.2. Antioxidant Activity

Oxygen is essential for aerobic life [126]; however, it is a potential hazard because it promotes the formation of reactive oxygen species (ROS) [127]. An antioxidant is a

substance that delays or prevents the oxidation of a substrate. In addition, in the human body they stabilize the generated radical and reduce the oxidative damage [128].

The second bioactivity most evaluated was the antioxidant activity, and some widely used methods include DPPH [82], ABTS [40,90,102], reducing power assays [92], and hydroxyl [67] and superoxide radical scavenging activity [94]. Other, less common methods, include the ferric thiocyanate method [88], lipid peroxidation, and the protein oxidation inhibition assay [91]. On the other hand, different kinds of results for antioxidant activity have been reported: $IC_{50}$ values and equivalence to vitamin C (see Table 7), percentage to a specific concentration (see Table 8), and some descriptive results (see Table 9).

**Table 7.** Results ($IC_{50}$ and equivalence to vitamin C) of the *Streptomyces* strains with antioxidant activity reported.

| *Streptomyces* Strain (Source) | Antioxidant Method Evaluated | $IC_{50}$ (μg/mL) | Equivalence to Vitamin C (μg) | Ref. |
|---|---|---|---|---|
| *Streptomyces* sp. A1013Y (TDTA (18)) | DPPH<br>ABTS | 41<br>14 | <1<br>1 | [82] |
| *Streptomyces glaucescens* KCTC988 (Melanin) | ABTS<br>ABTS (In presence of copper ions) | 25,080<br>7890 | N/A<br>N/A | [102] |

**Table 8.** Results (concentration evaluated and percentage) of the *Streptomyces* strains with antioxidant activity reported.

| *Streptomyces* Strain (Source) | Antioxidant Method Evaluated | Concentration Evaluated (μg/mL) | Percentage of Activity | Ref. |
|---|---|---|---|---|
| *Streptomyces* sp. (Melanin) | Hydroxyl radical scavenging activity | 500 | 70% | [67] |
| *Streptomyces glaucescens* NEAE-H (Melanin) | ABTS | 100 | 57% | [40] |
| *Streptomyces parvus* BSB49 (Eumelanin) | DPPH<br>ABTS | 250<br>250 | 88%<br>75% | [90] |
| *Streptomyces* sp. WMA-LM31 (Prodigiosin) | DPPH<br>Lipid peroxidation inhibition assay<br>In vitro protein oxidation inhibition assay | 10<br>10<br>10 | 60%<br>25%<br>55% | [91] |
| *Streptomyces* sp. ZL-24 (Melanin) | DPPH<br>Hydroxyl radical scavenging activity<br>Superoxide scavenging activity | 5<br>50<br>10<br>50 | 65%<br>96%<br>43%<br>60% | [94] |

**Table 9.** Descriptive results of the *Streptomyces* strains with antioxidant activity reported.

| *Streptomyces* Strain (Source) | Antioxidant Method Evaluated | Results of Antioxidant Activity | Ref. |
|---|---|---|---|
| *Streptomyces* sp. (Melanin) | Superoxide radical scavenging activity<br><br>Reducing power assay | Moderate scavenger of superoxide radical in vitro and exhibited a strong dose–effect relationship.<br>Antioxidant activity of melanin might be due to redox reactions. | [67] |
| *Streptomyces* sp. JS520 (Undecylprodigiosin (2)) | Ferric thiocyanate method<br><br>Hydrogen peroxide disc-diffusion assay | Undecylprodigiosin (2) did not perform as well as commercially available antioxidant α-tocopherol; however, it was effective in delaying lipid peroxidation.<br>Undecylprodigiosin (2) acted as a scavenger of $H_2O_2$ that is released through the process of peroxidation. | [88] |
| *Streptomyces* sp. JAR6 (Undecylprodigiosin (2)) | DPPH | Strain JAR6 was able to reduce compounds to pale yellow hydrazine as a DPPH radical. | [89] |
| *Streptomyces coelicolor* MSIS1 (Extract) | Reducing Power Assay | The pigment had positive results for all the concentrations: 10 mg/mL, 50 mg/mL, and 100 mg/mL. | [92] |
| *Streptomyces cavourensis* SV 21 (Melanin) | DPPH<br><br>Hydroxyl radical scavenging activity | Acid-treated forms of melanin showed much stronger radical scavenging ability than the intact melanin derivatives.<br>Rapid oxidation and bleaching of the melanin pigment and thus its capacity to scavenge $H_2O_2$ out of the environment. | [70] |
| *Streptomyces* sp. (Extract) | Free radical scavenging activity Ferric Reducing Antioxidant Power Hydroxyl Radical Scavenging Activity | The pigment showed increasing free radical scavenging activity and total antioxidant activity with increased concentrations. | [68] |

Two of the most used methods for the evaluation of antioxidant activity are DPPH and ABTS assays. In the cases in which the antioxidant capacity of the same sample was evaluated using these two methods (see Tables 4 and 5), it was evident that there was a difference; the DPPH assay is applicable to only hydrophobic systems [129] while the ABTS assay is applicable to both hydrophilic and lipophilic. Additionally, it is suggested that the ABTS assay better reflects the antioxidant contents than the DPPH assay [130] and is considered to be a method of high sensitivity, which is practical, fast, and very stable [131].

Most reported *Streptomyces* pigments show high antioxidant activity, regardless of the evaluation method or the units of reporting, demonstrating that these pigmented extracts or compounds are promising as antioxidant agents, with potential applications in the cosmetic industry. Additionally, melanin is highlighted as an antioxidant which, regardless of its origin, has already been reported to have other self-protective roles in response to elevated environmental stress conditions, such as antiultraviolet radiation, chelating metal ions, high temperature tolerance [94,132], and bioactivities (e.g., antibiotic and anticancer) [70].

2.6.3. Cytotoxic Activity

Cancer is a major public health threat worldwide as the leading cause of morbidity and mortality [89,93,133,134]. Even worse, cancer treatments such as chemotherapy, surgery, and radiotherapy are unsatisfactory. This is due to the high complexity of the disease and its wide variety of molecular mechanisms for attacking cancer cells, the rapid evolution of resistance to today's multiple anticancer drugs, and the drug side effects [84,85]. For

these reasons, searching for new secondary metabolites for cancer treatment that are more effective and safer is an urgent priority [84,85,89,93].

In the articles included in this review, for the evaluation of the cytotoxic activity, were used cancer cell lines, such as fibro sarcoma (HT1080), larynx (Hep2), cervical (HeLa), breast (MCF7), liver (HepG2), skin (HFB4), human carcinoma of nasopharynx cell (KB cells); and noncancer cell lines, such as human lymphocytes, peripheral blood mononuclear cells (PBMCs), human lung fibroblast (WI-38), human amnion (WISH), human epidermal keratinocyte (HaCat), fetal lung fibroblast (MRC-5), human embryonic kidney (HEK293), and human melanoma (SK-MEL-28). In most cases, cytotoxic activity is reported using the half-maximal inhibitory concentration ($IC_{50}$) in concentration units (μg/Ml), which is the amount of a specific drug needed to inhibit a biological process by half [135] (see Table 10).

**Table 10.** $IC_{50}$ (μg/mL) of *Streptomyces*-derived pigments and their conditions (cell lines and cell density, among others).

| *Streptomyces* Strain | Pigment | Concentration (μg/mL) | Cell Line | Cell Density (Cells/Well) | Time | IC50 (μg/mL) | Ref. |
|---|---|---|---|---|---|---|---|
| *Streptomyces* sp. PM4 | Red pigment | 10, 20, 30, 40, 50 | HT1080 Hep2 HeLa MCF7 | $2 \times 10^4$ | 24 h | 18.5 15.3 9.6 8.5 | [61] |
| *Streptomyces griseoaurantiacus* JUACT 01 | Yellow Pigment | 2.5, 5, 10, 20 | HeLa HepG2 Human lymphocytes | N/A | 24 h 48 h 72 h 24 h 48 h 72 h 24 h, 48 h, 72 h | 5.31 2 1.8 26.33 1.75 1.41 Any cytotoxicity | [84] |
| *Streptomyces* sp. A 16-1 | Red pigment (Fr 5, Fr6, and Fr7) | 0–8 | KB cells PBMCs | $5 \times 10^4$ | 48 h | $0.04 \pm 0.005$ (Fr 5) $0.20 \pm 0.02$ (Fr 6) $0.55 \pm 0.05$ (Fr 7) Low Cytotoxicity | [85] |
| *Streptomyces* sp. JAR6 | Red pigment (Undecylprodigiosin (2)) | 18.75, 37.5, 75, 150, 300 | HeLa | $1 \times 10^4$ | 48 h | 145 | [89] |
| *Streptomyces glaucescens* NEAE-H | Melanin | 1.56, 3.125, 6.25, 12.5, 25, 50, 100 | HFB4 WI-38 WISH | $1 \times 10^4$ | 24 h | $16.34 \pm 1.31$ $37.05 \pm 2.40$ $48.07 \pm 2.76$ | [40] |
| *Streptomyces* sp. WMA-LM31 | Prodigiosin (4) | 5, 10, 15, 20 | HepG2 HeLa | $1 \times 10^4$ | 24 h | 12.66 14.83 | [91] |
| *Streptomyces parvus* BSB49 | Eumelanin | $2.72 \times 10^6$–$1.09 \times 10^{7}$ [1] | HeLa | $3 \times 10^4$ | 24 h | $5.45 \times 10^{6}$ [1] | [90] |
| *Streptomyces* sp. NP4 | Prodigiosin (4) | N/A [2] | HaCat MRC-5 | $1 \times 10^4$ | 48 h | No significant cytotoxic effect | [83] |
| *Streptomyces* sp. | N/A | N/A | HT-1080 HeLa | N/A | N/A | 202.13 253.86 | [68] |

[1] The original units were mM and the conversion was performed using the molecular weights from pubchem.com.
[2] The concentrations unit were different: 12.5%, 25%, 50%, and 100% (*v/v*).

Red pigments showed promise as anticancer agents. Extracts of *Streptomyces* sp. PM4 strain [61] and fractions of *Streptomyces* sp. A 16-1 [85] extract showed activity at very low concentrations in the range of 0.04–18.5 μg/mL ($IC_{50}$ value) against cancer lines and are harmless against healthy lines such as PBMCs. Specifically, red compounds such as undecylprodigiosin (2) showed activity against HeLa [89], and prodigiosin (4) is safe against healthy cell lines [83].

On the other hand, the yellow-pigmented extract was tested against cell lines for different exposure times, showing that the longer the exposure time, the lower the $IC_{50}$

value. Again, it is safe against a healthy human lymphocyte line [84]. Likewise, melanin from different strains showed great cytotoxic activity against different cancer lines, but its toxicity against healthy lines was predominant [40].

In other cases, cytotoxic activity was reported using different measurements such as growth inhibitory activity ($GI_{50}$), which is the concentration of the evaluated compound required to cause a 50% decrease in net cell growth [136]; and lethal concentration 50 ($LC_{50}$), which is the concentration of a given agent that obtains a cellular lethality of 50% (see Table 11) [137].

**Table 11.** $GI_{50}$ and $LC_{50}$ of *Streptomyces*-derived pigments and their conditions (cell lines and cell density, among others).

| *Streptomyces* Strain | Pigment Evaluated | Concentration (µg/mL) | Cell Line | Time of Treatment | $GI_{50}$ (µg/mL) | $LC_{50}$ (µg/mL) | Ref. |
|---|---|---|---|---|---|---|---|
| *Streptomyces aurantiacus* AAA5 | Resistomycin (19) | 5, 0.5, 0.05, 0.005, 0.0005 | HepG2 HeLa | N/A | $5 \times 10^{-3}$ $6 \times 10^{-3}$ | $1 \times 10^{-2}$ $1 \times 10^{-2}$ | [93] |

Resistomycin (19), besides being an excellent natural antibiotic, had its cytotoxic activity evaluated by Vijayabharathi et al. [93] in 2011. It was later studied in detail by Han et al. [138], who determined that resistomycin (19) activates the p38 MAPK signaling pathway, causing apoptosis and G2/M phase arrestin.

Another way to evaluate the cytotoxic activity is by in vivo assays, wherein the measurement reported is the median lethal dose ($LD_{50}$) (see Table 12) [137]. Only actinomycin X2 (10) and λ-Actinorhodin (12) pigment toxicities were evaluated in the brine shrimp *A. salina* and Mouse, respectively. In both cases, the pigment had a good biological safety property [69,86].

**Table 12.** $LD_{50}$ of *Streptomyces*-derived pigments and the in vivo conditions.

| *Streptomyces* Strain | Type of Assay | N° | Doses (mg/kg) | General Results | $LD_{50}$ (mg/kg) | Ref. |
|---|---|---|---|---|---|---|
| *S. coelicolor* 100 (λ-Actinorhodin (12)) | Mouse acute toxicity trial | First Assay | 1500 and 15,000 | Mouse death resulted from taking an overdose pigment once by oral gavages. | >15,000 | [86] |
|  |  | Second Assay | 0, 464, 1000, 2155, 4633, 10,000, and 15,000 | Nontoxic substance |  |  |

### 2.7. Applications of Streptomyces Pigments

*Streptomyces* pigments have a wide variety of applications, including their application as antimicrobial (26.1%), anticancer (17.4%), and antioxidant (10.1) agents. Additionally, 13% of the *Streptomyces* pigments do not have a specific application; therefore, they require further study and may have great biotechnological potential (Figure 16). For the specific application of each pigment, see material Supplementary Tables S7–S13.

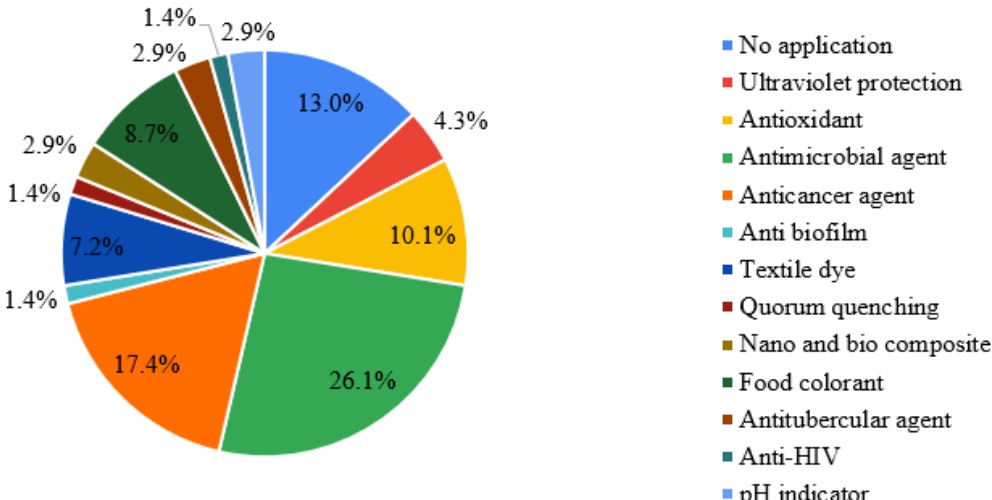

**Figure 16.** Main applications of *Streptomyces* pigments.

Especially, Wibowo et al. [70] studied the activity of purified dissolved melanin (PDM), acid-based precipitation of melanin (AM), and synthetic melanin standard (SM) against *Alivibrio fischeri*, determining that the quorum-sensing activity of *A. fischeri* was interrupted more clearly by PDM and SM. Additionally, that was the first report of this activity on melanin and it proposed that the melanin from *S. cavourensis SV 21* may have an important function for the microbe–host and/or microbe–microbe interaction. In the same way, Wang et al. [94] reported that insoluble and soluble melanin pigments could reduce biofilm formation against the Gram-positive *M. smegmatis* ATCC 10231 and the Gram-negative *P. aeruginosa* ATCC 9027 in a dose-dependent manner.

On the other hand, the pink pigment of *Streptomyces* sp. NS-05 was used to synthesize silver nanoparticles (AgNPs) which showed antimicrobial activity against Gram-positive and Gram-negative bacteria and can be used for the green synthesis of other nanoparticles [81]. In addition, Bayram [107] determined that the amorphous organic semiconductor, X-ray, and γ-ray-absorbing properties of pyomelanin polymers require more investigation for use in nanocomposite and biocomposite material production.

Other notable applications are as antituberculars and anti-HIVs. Thus, the pinkish-brown-pigment-producing *Streptomyces* sp. S45 showed anti-HIV activity with the IC50 value of 8.75 μg/mL [63]. In addition, the pigment from *Streptomyces* sp. SFA5 was evaluated against M. tuberculosis H37Rv and for inhibitory activity against *M. tuberculosis* lysine aminotransferase, showing activity in both assays and an $IC_{50}$ value of 4.5 μg/mL concentration for the last one [79].

Interestingly, some of the elucidated pigments were not determined to have a potential application or the only bioactivity evaluated gave negative results; therefore, their potential application could not be determined. Some examples of these facts are streptorubin A and B (3) [64], grixazones A (14) and B (15) [98], and naphthoquinone derivatives (5–7) [72] (see Table S7). Nevertheless, melanin from different *Streptomyces* strains has a wide repertoire of potential applications, including antioxidant, photoprotection, antimicrobial, antibiofilm, quorum-quenching inhibitor, textile dye, anticancer and anti-inflammatory [70,94]. Likewise, its analogue eumelanin has some of its applications (photoprotection, antioxidant, and anticancer) [90,110] (see Table S8).

Especially, the actinomycin family includes actinomycin $L_1$ (8) and $L_2$ (9) [80] and Actinomycin X2 (10) [69], which are highlighted as antimicrobial agents, and the latter also has potential application as a textile dye (see Table S9). Compounds of the actinorhodin family (γ-Actinorhodin (11), λ-Actinorhodin (12), and Actinorhodin (13)) from different strains of *Streptomyces coelicolor* have potential applications as food colorants or antimicrobial agents [86,101,103,104,109]. Additionally, compounds of the prodigiosin family (undecylprodigiosin (2), metacycloprodigiosin (1), and prodigiosin (4)) have a broad spec-

trum of applications, including food and textile dyes, and as antimicrobial, anticancer, and antioxidant agents [77,83,88,89,91,99,106] (see Table S10).

*Streptomyces* Pigments with Antibiofilm/Antifouling Potential

　　Bacterial biofilms have a structural complex architecture and develop on many abiotic surfaces (plastic, glass, metal, and minerals) and biotic surfaces (plants, animals, and humans) [139]. Bacteria growing on biofilms is up to 1000-fold more resistant to antibiotics and biocides compared to their planktonic counterparts [140]. Biofilms are the root cause of biofouling [141]. Specifically, the attachment of micro- and macroorganisms to water-immersed surfaces is an undesired phenomenon in some cases, and is known as marine biofouling, resulting in severe problems for aquaculture, shipping, and other industries that rely on coastal and off-shore infrastructures [142].

　　Some publications described the potential of *Streptomyces* to produce antibiofilm/antifouling pigmented extracts and compounds that are active against micro- and macrofouling organisms [142–146], especially founded napyradiomycin (20–31) and flavonoids. The first are highly reactive organic compounds used as dyes, whose colors range from yellow to red [112]. The second are natural dyes used as mordant, and among these is quercetin (33), which is part of flavonol, one of the main chromophores in flavonoids with a yellow color; and one of its derivatives, the flavonol taxifolin (32) or dihydroquercetin [147].

　　Napyradiomycin (naphthalene quinone) derivatives (Figure 17) that were isolated from *Streptomyces* from ocean sediments from the Madeira Archipelago presented antifouling activity. Pereira et al. (2020) [142], revealed that napyradiomycins (20–31) inhibited ≥80% of the marine-biofilm-forming bacteria assayed, as well as the settlement of *Mytilus galloprovincialis* larvae. Napyradiomycin derivates (20–31) disclosed bioactivity against marine micro- and macrofouling organisms and nontoxic effects towards the studied species, displaying potential to be used in the development of antifouling products [142].

|    | $R_1$  | $R_2$     |
|----|--------|-----------|
| 20 | H      | $CH_3$    |
| 21 | H      | $CH_2OH$  |
| 22 | $CH_3$ | $CH_3$    |

|    | $R_1$ | $R_2$ |
|----|-------|-------|
| 23 | H     | OH    |
| 24 | =O    | =O    |

**Figure 17.** *Cont.*

**Figure 17.** Chemical structures of napyradiomycins isolated from marine-derived *S. aculeolatus* strains PTM-029 [142].

Gopikrishnan et al. (2019) [145], reported the isolation, characterization, and potential antifouling activity of taxifolin (32), a flavonoid compound from *Streptomyces* sp. PM33 isolated from mangrove sediments (Figure 18). Toxicity assays based on zebra fish models revealed the less or moderate toxicity of this metabolite. Taxifolin (32) showed significant potential to fight against biofilm formation. It inhibited algal spore germination and mollusk foot adherences were the main mechanism of antibiofilm activities of the metabolite. Taxifolin (32) in the field experiments revealed good antifouling activity when tested on wooden surfaces and PVC panels [145].

Taxofolin (32)

**Figure 18.** Chemical structures of taxifolin.

Sheir and Hafez, in 2017 [148], demonstrated that *S. toxytricini* fz94 crude pigmented extract was an effective and safe anti-*Candida* biofilm at concentrations in prevention and destruction modes. It was similar to ketoconazole against clinical *Candida* isolates and it was more potent than ketoconazole in the destruction of *C.albicans* biofilm [148].

Gopikrishnan et al., in 2016 [149], reported on quercetin (33) from marine-derived *Streptomyces* sp. PE7 with antibiofouling activity (Figure 19). It was active against 18 biofouling bacteria with an MIC range between 1.6 and 25 µg/mL and had algal spore germination and mollusk foot adherence found at 100 µg/mL and $306 \pm 19.6$ µg mL$^{-1}$, respectively. Previous research by the same research group [146] obtained a crude pigment from the *Streptomyces* sp D25, produced by agar surface fermentation using yeast extract and malt extract agar and extracted using ethyl acetate. The pigmented extract exhibited antioxidant potential in DPPH and nitric oxide assays and antimicrobial activity against the biofilm-forming bacteria in the disc-diffusion method. Further in vivo studies on this *Streptomyces* pigment pave the way for its biomedical applications. With the above-mentioned research, it is possible to propose the use of pigmented extracts or metabolites of *Streptomyces* that are ecofriendly and can be the basis for the preparation of biocidal paints to coat different

surfaces and thus protect them from biofilm or biofouling attack, which could replace the available chemical preparations with antibiofilm or antifouling potential.

Quercetin (33)

**Figure 19.** Chemical structures of quercetin.

To our knowledge, most research has evaluated the pigmented compound against biofilm-forming microorganisms; however, studies with Streptomyces pigments have been limited only to biocidal activity and quorum sensing inhibitors, two fundamental bioactivities in biofouling. Future research is required to evaluate these Streptomyces pigments that have already demonstrated these bioactivities in antifouling assays, and this may be the beginning to expand the repertoire of candidates useful in the development of surface coatings that need to be protected from the complex bioprocess of biofouling.

*2.8. Future Perspectives*

Only 50.9% of the articles reported partial or complete purification, 13% had no determined application, and others had potential applications without specific tests and poorly studied isolation sources, such as freshwater (Free-living) or terrestrial symbionts. Future research includes purifying pigmented extracts that have already been reported, studying the purified compounds in a specific application, and isolating new microorganisms from new or poorly studied isolation sources.

However, research has also focused on improving the production of pigments already identified with applications, using different optimization methods to achieve large-scale production, changing the culture media, the main sources of nutrients, and their culture conditions. In addition, taking into account the economy, the large amount of agroindustrial waste, and the Sustainable Development Goals Fund, the use of new sources of nutrients, such as sugar cane waste, rice bran, wheat bran, coconut cake, and rice flour, is a new subject of study (as performed by Vasanthabharathi et al. [60]).

Not only the culture media have been the subject of study, but also the use of new technologies such as solid-state fermentation (SSF) using a novel PolyHIPE Polymer (PHP) matrix in a microbioreactor for improving the production of antibiotics from *S. coelicolor* A3(2) [101]. In other ways, it is required to recognize how conditions influence product formation, and online monitoring emerges as a great and valuable tool. For example, Finger et al. [109] determined that oxygen transfer rate and autofluorescence are key features in understanding the cultivation of the model organism *Streptomyces coelicolor* A3(2) [109].

On the other hand, pigment production has focused on downstream processes, improving existing extraction techniques or developing new techniques, including vacuum-freeze-drying or lyophilization [48,49,73], flash evaporation [60,108] or pressure reduction [61], ultrasonic cell-disrupter system to perform lysis [67] or sonication [91], and filtration using membranes [98] and ultrafiltration membranes [100]. Likewise, a wide range of ecofriendly and efficient solvents have been studied and the extraction processes improved using solvents in different ratio [63], using different solvent systems [84], or performing a re-

extraction many times [61,82]. As an example of this, Wibowo et al. [70] studied the two forms of melanin from *Streptomyces cavourensis* SV 21 and proposed a novel acid-free purification protocol of purified particulate melanin (PPM) and purified dissolved melanin (PDM) [70].

Finally, considering the great variety of Streptomyces pigment bioactivities (antioxidant, antimicrobial, and cytotoxic), the great variety of colorations, and the relative safety against healthy cell lines, Streptomyces pigments may be a valuable biotechnological resource with potential applications as nutraceuticals and a potential replacement for synthetic dyes. Additionally, it is possible to propose the use of pigments of Streptomyces that are ecofriendly and can be the basis for the preparation of biocidal paints to coat different surfaces and thus protect them from biofilm or biofouling attacks. Further studies of Streptomyces pigments will be necessary to optimize the bioprocesses and scale-up production, as will preclinical studies and in situ trials to fully establish their feasibility.

## 3. Materials and Methods

### 3.1. Databases and Search Strategy

For a review of the literature as complete as possible, the search was performed using the following databases: Scopus, Web of Science, and PubMed. The terms (including synonyms and related words) and boolean operators used for all searching were defined as follows:

Streptomyces AND (pigment OR colorant OR stain OR dye OR coloring OR tint).

### 3.2. Selection Procedure

The selection of the articles was based on the following inclusion criteria: (a) original research articles; (b) studies on extracts, compounds, fractions, or pure cultures derived from *Streptomyces* strains; and (c) studies related to pigment production (evaluation of bioactivities, purification or/and elucidation, and optimization of the production).

The following were considered exclusion criteria: (a) articles were written in a language other than English, and (b) articles whose full-text version could not be accessed.

The article selection process was subdivided into two stages as follows: in the first stage, four researchers separately assessed each title and abstract in a blind process. At this time, each article was marked as potentially eligible to be included in the review when at least two studies indicated that it met the inclusion/exclusion criteria. When an article was indicated as eligible by only one researcher, a discussion within the research team was carried out to solve disagreement. In the second stage, potentially eligible articles were examined at the full-text level. Thus, those articles that complied with the inclusion/exclusion criteria were finally selected for data extraction.

### 3.3. Data Collection and Tabulation

To guarantee careful and cautious data collection and avoid the risk of bias, a pilot data acquisition form was prepared. The form was evaluated and improved through an exercise including ten randomly selected articles. In this manner, having defined the final version, the form was used for the data acquisition of the complete set of selected articles.

**Supplementary Materials:** The following supporting information can be downloaded at: https://www.mdpi.com/article/10.3390/coatings12121858/s1, Table S1. PRISMA checklist. Table S2. Gram-negative Bacteria evaluated in the antimicrobial test. Table S3. Gram-positive Bacteria evaluated in the antimicrobial test. Table S4. Mushrooms and Yeast evaluated in the antimicrobial test. Table S5. *Streptomyces* strains source of bioactive crude extracts. Table S6. List of compounds retrieved from the included papers. Table S7. *Streptomyces* pigments without any specific application. Table S8. *Streptomyces* pigments belonging to the melanin family. Table S9. *Streptomyces* pigments belonging to the actinomycin family. Table S10. *Streptomyces* pigments belonging to the actinorhodin and prodigiosin family. Table S11. Yellow *Streptomyces* pigments and its applications. Table S12. Red and pink *Streptomyces* pigments and its applications. Table S13. Other *Streptomyces* pigments and

its applications. Molecules from compounds retrieved from the included papers. Table S14. List of molecules retrieved from the included papers.

**Author Contributions:** Conceptualization, L.D. and J.S.-S.; methodology, A.A.S.-T., L.S., J.S.-S. and L.D.; software, A.A.S.-T.; validation, A.A.S.-T.; formal analysis, A.A.S.-T. and L.S.; investigation, A.A.S.-T., L.S. and J.S.-S.; resources, L.D.; data curation, A.A.S.-T. and L.S.; writing—original draft preparation, A.A.S.-T., L.S. and J.S.-S.; writing—review and editing, A.A.S.-T., L.S., J.S.-S. and L.D.; visualization, A.A.S.-T.; supervision, J.S.-S. and L.D.; project administration, L.D.; funding acquisition, L.D. All authors have read and agreed to the published version of the manuscript.

**Funding:** This research was funded by Universidad de La Sabana (General Research Directorate, project ING-204-2018).

**Institutional Review Board Statement:** Not applicable.

**Informed Consent Statement:** Not applicable.

**Data Availability Statement:** Data supporting the reported results can be found in this document and in the Supplementary Materials. If they become required, please request them by mail at luis.diaz1@unisabana.edu.co.

**Acknowledgments:** We acknowledge the Universidad de La Sabana (General Research Directorate), the GIBP and Actinos Group for their support, especially to Maria Clara De La Hoz-Romo.

**Conflicts of Interest:** The authors declare no conflict of interest. The funders had no role in the design of the study; in the collection, analyses, or interpretation of data; in the writing of the manuscript; or in the decision to publish the results.

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
