# Peer review of "Streptomyces-Derived Bioactive Pigments: Ecofriendly Source of Bioactive Compounds"

_coatings, doi:10.3390/coatings12121858_

Round 1
Reviewer 1 Report (New Reviewer)
A document with the reviewer's comments is attached.

Author Response
Dear Reviewer thanks a lot for your time and your comments and suggestions. Please see the attachment.

Reviewer 2 Report (New Reviewer)
The manuscript has given a systematic review on the progress in pigment production from Streptomyces spp. The manuscript is interesting and using PRISMA is appreciable. However, a minor revisions will be required before publishing the manuscript.
1. In some places the organism name is not italized.
2. In table 2, whether the study tested cells in disk diffusion method or with crude extract or pigments against different pathogens is not clear. If it is pigment, then mention the name of the pigment and the state in which it is tested (crude, purified, etc.,) which will be helpful.
3. In case of table 3, the state of pigment in which it is tested (crude, purified, etc.,) which will be helpful.
4. I cant understand why the authors have given the bioactivity results in the first section. If possible, this section can be kept before applications.
5. The chemical structure figures are little distorted. A high resolution pics from chemdraw can be included.
Author Response
Please see the attachment.

Reviewer 3 Report (New Reviewer)
The review article by Sarmiento-Tovar et al. summarizes the biological activities of pigments isolated from bacteria belonging to the genus Streptomyces and it contains an important number of citing literature.
My main preoccupation is whether the article fits to the scope of the journal Coatings. The discussion must be extended on the existing or potential applications of these natural pigments in the field of nanotechnology, material science, surface modification, for example for the preparation of functional films, coatings, composites, etc. Otherwise, this review article could be published in another journal related to natural products.
I recommend the publication in Coatings after major revision and addressing of the issue mentioned above, as well as some presentation problems.
Thereofore, I suggest the following changes:
1) In the Introduction, it is written (line 65, page 2) „the use of natural pigment in food, paints, cosmetics, and pharmaceuticals has increased in recent decades“, and (line 77, page 2) „revealing its potential for various industrial applications“ without mentioning examples of these potential applications. For example, some advanced applications in preparation of antimicrobial and photoprotective can be envisaged. Antifouling surfaces are mentioned in the references 13, 27, 28 but not explained in the text.
In this line, the Section 2.7: Applications of Streptomyces Pigments (page 23) can be divided in subsections to discuss the therapeutical potential of these compounds and separately the potential in material science including the references 58 and 93, if possible expand this part.
2) Are older reviews that deal with the same subject? Please mention them and explain what it is different in your review.
3) Each compound when it is mentioned either in the text or in the tables should have a number that corresponds to a structure. The names and numbers should be followed from the beginning to the end of the document and not only in the biosynthesis part. It is easier for the reader to have a table or a figure/figures with the structures of the compounds, their number, and perhaps their isolation sources.
For the different melanins, perhaps a core structure may be given.
4) Which are the compounds in the Table 1?
5) In the Tables with the activities, it is useful to specify whether the activity is due to an extract or the whole culture.
Round 2
Reviewer 1 Report (New Reviewer)
The manuscript presents an interesting review related to the biological activities and applications of pigments derived from Streptomyces (an industrially important bacterium). It presents the analysis of the literature of about 50 years. Furthermore, it focuses on the various pigments produced by this genus, and not only on one type of pigment.
The manuscript is presented with details of the literature findings and contains information that is relevant to the area related to microbial pigments. The systematic analysis provides insight into the advances and highlights the research needs required to increase knowledge in this field of study and application.
The authors have made the pertinent changes suggested by the reviewers and have made a more appropriate organization of the paper. The manuscript is recommended, provided that some minor revisions, related to the following, are considered:
1. It is suggested that the information in the tables be standardized in terms of presentation, such as the presentation of numerical data. In some tables, they are presented with decimal values and in others not, in addition, some include standard deviations and others do not. The use of notes in the table, as presented in some, can be useful in the rest.
2. In Table 1, it is suggested to indicate in column 4 (No. Compounds), where the value of zero (0) is presented, if pure compounds are not described, or if they are extracts and for that reason zero is indicated. This is to avoid confusion.
3. Line 405 to 407, it is suggested that this paragraph be introduced somewhere in the paragraphs in lines 421-430. In this way, there can be continuity in the presentation of the information, which can be more congruent with what is mentioned in the paragraphs on lines 421 to 430, instead of the paragraph where it is currently located.
4. Spacing needs to be revised in some sections of the document, for example, line 431.
5. A new section (2.7.1) is described on lines 679-724. The subject matter is interesting to discuss, however, a revision of this section is recommended, in the sense of making clear the role of Streptomyces-derived pigments exhibiting these activities. For example, indicate whether such activities are suspected to be due to the pigments or whether they may be due to other related molecules. As well as to establish whether little information exists in this regard.
Author Response
Please see the attachment

Reviewer 3 Report (New Reviewer)
The authors have significantly improved their manuscript. I recommend its publication in Coatings after a minor revision: addition of the chemical structures and numbers in the text of the new compounds mentioned in the Section "2.7.1 Streptomyces Pigments with antibiofilm/antifouling potential".
Author Response
Please see the attachment

This manuscript is a resubmission of an earlier submission. The following is a list of the peer review reports and author responses from that submission.
Round 1
Reviewer 1 Report
The present review aimed to introduce Streptomyces-derived bioactive pigments. However, no chemical structure of these pigments or structure-activity relationship was decribed. It extensively focused on pigmented extracts and their biological properties. Therefore, the conception and organization of this work is unsuitable for the journal and should be revised.
Author Response
Dear Reviewer
We would like to thank you for taking the time to review the manuscript and for your appreciation of our work. We deeply regret that you were not completely satisfied with our manuscript.
We wish to clarify that within our results we found that the number of articles that perform a partial or complete purification are almost the same as the extracts and even fewer are those that elucidate the chemical structure. Due to the above, in the future perspectives section we highlight: "future research includes purifying pigmented extracts that have already been reported".
The few studies that have determined the chemical structure derive in 22 molecules found in the supplementary material (Table S6 and S14), this reduced group corresponds to a wide variety of structures, which does not allow grouping them by structured cores and at the same time the association of these cores with the reported bioactivities is not clear.
Likewise, based on the supplementary material (Table S7-S13), we grouped by family in the case of the compounds and by coloration in the case of the extracts; noting that the same family has a great variety of bioactivities and potential applications, which does not allow us to make a classification relating structure-bioactivity; we did not find a common pattern or nucleus due to the wide structural variety of the pigments that present the same bioactivity.
We value your feedback and we would be honored if you would give our manuscript a new opportunity and could evaluate it from a new perspective with the changes and clarifications made.
Reviewer 2 Report
The manuscript entitled "Streptomyces-Derived Bioactive Pigments: A Systematic 2 Literature Review", discusses the latest advances of pigments-producing Streptomyces in terms of identification, purifications strains, biological activities and future perspectives. The manuscript is very well- designed, developed and written. It represents a good-step forward that enriched the literature of of applications of Streptomyces. I'd like to thank the authors for this nice piece of work and wish them all the best.
Author Response
We would like to thank you for taking the time to review the manuscript and for your appreciation of our work. It is very valuable to us as authors that you have found our manuscript of great value to enrich the Streptomyces applications literature.
Thank you very much for these best wishes.
Reviewer 3 Report
Dear editor, the manuscript is well written and presented. Here are some suggestions about the review article.
1. Authors need to focus some more on the drawbacks of synthetic pigments in one section along with the advantages of natural pigments, especially Streptomyces-Derived Bioactive Pigments in another section.
2. Applications are not listed in the review article for pigments produced like naphthoquinone derivatives, blue, red λ-actinorhodin etc. all pigments that listed in table 11. and could be added to Section 2.6. Applications of Streptomyces Pigments," as it is not mentioned
3. the rest of review article are well written
Author Response
Dear Reviewer
We would like to thank you for taking the time to review the manuscript and for your appreciation of our work. Your recommendations are very valuable as authors and relevant to our manuscript. In response to your suggestions, we have made the following changes:
- “Authors need to focus some more on the drawbacks of synthetic pigments in one section along with the advantages of natural pigments, especially Streptomyces-Derived Bioactive Pigments in another section”
We added a paragraph in the introduction (lines 43-56) focusing on the disadvantages of synthetic pigments and some examples of widely used synthetic pigments with some contraindications. Additionally, we highlight in the “future perspectives” section (lines 491-495) the results of Streptomyces pigments as a valuable resource and promising replacement for synthetic pigments.
- “Applications are not listed in the review article for pigments produced like naphthoquinone derivatives, blue, red λ-actinorhodin etc. all pigments that listed in table 11. and could be added to Section 2.6. Applications of Streptomyces Pigments," as it is not mentioned”
In the supplementary material (Table S7-S13) are found the possible applications of the pigments (extracts and compounds) grouped by family or coloration to avoid making the manuscript too long.
However, we understand the recommendation and therefore added two paragraph in this section (lines 441-457) highlighting the applications mainly of the identified compounds.
Again, we thank you for your recommendations and comments; and we are pleased to receive your feedback on the new version of the manuscript.
Round 2
Reviewer 1 Report
As one of important journals in the field of chemistry, chemical structure of each molecule or substance should be unambiguously characterized and confirmed. Natural pigments consist of numerous compounds with various carbon skeletons, chromophores and biosynthetic pathways, and these points should be clearly elucidated in this review.
Author Response
Dear Reviewer
Newly, we would like to thank you for taking the time to review the manuscript and for your appreciation of our work. Considering the recommendations given, we add section 2.4 entitled: "Biosynthetic pathways and structure of Streptomyces pigments" (lines 361-517), in which we present the structures of the Streptomyces pigmented molecules found in this systematic review together with a description and what has been found of their biosynthetic pathways.
We value your feedback, and we hope that we have understood your suggestions and have been able to complement the information suggested by you.